# Spin-$s$ rational $Q$-system

Jue Hou[1], Yunfeng Jiang[1] and Rui-Dong Zhu[2]

**1** School of physics & Shing-Tung Yau Center, Southeast University,
Nanjing 211189, P. R. China
**2** Institute for Advanced Study & School of Physical Science and Technology,
Soochow University, Suzhou 215006, China

## Abstract

Bethe Ansatz equations for spin-$s$ Heisenberg spin chain with $s \geq 1$ are significantly more difficult to analyze than the spin-$\frac{1}{2}$ case, due to the presence of repeated roots. As a result, it is challenging to derive extra conditions for the Bethe roots to be physical and study the related completeness problem. In this paper, we propose the rational $Q$-system for the XXX$_s$ spin chain. Solutions of the proposed $Q$-system give all and only physical solutions of the Bethe Ansatz equations required by completeness. This is checked numerically and proved rigorously. The rational $Q$-system is equivalent to the requirement that the solution and the corresponding dual solution of the $TQ$-relation are both polynomials, which we prove rigorously. Based on this analysis, we propose the extra conditions for solutions of the XXX$_s$ Bethe Ansatz equations to be physical.

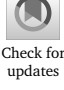

# 1  Introduction

The study of Bethe Ansatz equations (BAE) has a history that is as long as Bethe Ansatz itself. Essentially, Bethe Ansatz trades the problem of diagonalizing a big matrix (the Hamiltonian) to that of solving a set of algebraic or transcendental equations (the BAE). It is therefore not surprising that the BAE encodes rich information about the integrable model. Understanding solutions of BAE is a fundamental task for integrability. Developing efficient methods for solving BAE in different regimes is a crucial step in almost any applications of integrable models. At the same time, the algebraic beauty and rich structure have made the study of BAE a subject of considerable mathematical interest.

Most of studies in the past have focused on the spin-$\frac{1}{2}$ Heisenberg XXX and XXZ spin chain [1–9]. Thanks to these efforts, we now have a much deeper understanding about the solutions of the BAEs. The progress is largely facilitated by considering alternative formulations of BAE, such as Baxter's $TQ$-relation [10] and the rational $Q$-system [11–16], which was based on various developments in the analytic $TQ$ and $QQ$-systems in the past few decades [17–22]. Before turning to the BAE of $XXX_s$ spin chain, let us recap the situation in the spin-$\frac{1}{2}$ case.

It is now well understood that directly solving BAE of the form

$$\left(\frac{u_k + \frac{i}{2}}{u_k - \frac{i}{2}}\right)^L = \prod_{j \neq k}^{M} \frac{u_k - u_j + i}{u_k - u_j - i}, \quad k = 1, \ldots, M, \tag{1}$$

in general gives too many solutions.[1] There are two kinds of solutions that are non-physical, *i.e.* the corresponding Bethe vectors are not eigenstates of the Hamiltonian. The first kind are the solutions containing repeated roots; The second kind are the non-physical singular solutions. The singular solutions are the ones containing an exact Bethe string of length 2 and take the following form $\{-\frac{i}{2}, \frac{i}{2}, u_3, \ldots, u_M\}$. If the roots $\{u_3, \ldots, u_M\}$ satisfy the additional condition

$$\prod_{j=3}^{M}\left(\frac{u_k + \frac{i}{2}}{u_k - \frac{i}{2}}\right)^L = (-1)^L, \tag{2}$$

the singular solution is physical, otherwise, it is non-physical and should be discarded. It has been tested numerically that if these two kinds of solutions are excluded, one obtains the correct number of physical solutions required by the completeness of Bethe Ansatz [23].

The relations of these results to the $TQ$-relation and rational $Q$-system are as follows. Baxter's $TQ$-relation reads

$$\tau(u)Q(u) = \left(u + \tfrac{i}{2}\right)^L Q(u - i) + \left(u - \tfrac{i}{2}\right)^L Q(u + i), \tag{3}$$

where $\tau(u), Q(u)$ are polynomials. The zeros of $Q(u)$ are the Bethe roots $\{u_k\}$. Taking $u = u_k$ in the $TQ$-relation, we recover BAE. However, this does not mean $TQ$-relation is completely equivalent to BAE. When there are repeated roots in $\{u_k\}$, the fact that $\tau(u)$ is a polynomial

---

[1]Here we only consider solutions where $\{u_k\}$ are finite. Roots at infinity correspond to $SU(2)$ descendant states. This is well understood and can be taken into account easily when counting the number of all physical solutions.

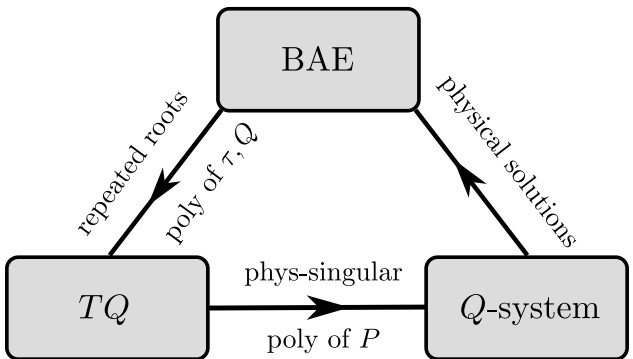

Figure 1: Relations between BAE, $TQ$ and rational $Q$-system. Polynomiality of $\tau(u)$ and $Q(u)$ imposes extra conditions for the repeated roots, which eliminates them in the spin-$\frac{1}{2}$ case; Furthermore, polynomiality of the dual solution $P(u)$ imposes extra conditions for physical singular solutions. The polynomiality of $\tau(u), Q(u)$ and $P(u)$ in the $TQ$-relation is equivalent to the rational $Q$-system.

imposes extra constraints for the Bethe roots. If we work directly with BAE, these extra constraints can be derived from the framework of algebraic Bethe Ansatz (ABA), see [24, 25]. It turns out that for spin-$\frac{1}{2}$ chain these extra conditions are not compatible with BAE.[2] As a result, we can discard the repeated roots. This implies that if we solve the $TQ$-relation instead of the BAE, the repeated roots are automatically eliminated in the spin-$\frac{1}{2}$ case.

The $TQ$-relation, however, does not eliminate non-physical singular solutions. For this, we need the rational $Q$-system. This method was first proposed as an efficient way to solve BAE [11]. Surprisingly, it leads to only physical solutions and eliminates also the non-physical singular solutions automatically. How the rational $Q$-system achieves this was somewhat mysterious at the beginning. Later it was demystified in [12] which also clarified several important points.

The crucial insight is that the rational $Q$-system is intimately related to the dual solution of the $TQ$-relation. Since $TQ$-relation (3) is a second-order difference equation. For a given solution $Q(u)$, there is a dual solution which we denote by $P(u)$ [26]. An important point is that even $\tau(u)$ and $Q(u)$ are polynomials, $P(u)$ is in general *not* a polynomial when the zeros of $Q(u)$ correspond to singular solutions. Requiring $P(u)$ to be a polynomial imposes extra conditions for $Q(u)$, which turns out to be precisely (2). One can prove that the rational $Q$-system is nothing but an ingenious reformulation for the requirement that $\tau(u)$, $Q(u)$, and $P(u)$ are polynomials. At the same time, it has been proven rigorously that the solutions of the $TQ$-relation with both $Q(u)$ and $P(u)$ being polynomials give the complete physical solutions for the corresponding BAE [5]. This explains why the rational $Q$-system works. The relations between BAE, $TQ$, and rational $Q$-system can be summarized in Figure 1.

Given the success of the spin-$\frac{1}{2}$ case, it is a natural next step to extend the insights and techniques to the more general XXX$_s$ spin chain. However, such an attempt meets an immediate difficulty: *repeated roots are allowed in the XXX$_s$ chain.* This simple fact complicates the analysis considerably. To start with, a generic non-singular solutions can take the following form

$$\{\underbrace{u_1,\ldots,u_1}_{n_1},\underbrace{u_2,\ldots,u_2}_{n_2},\ldots,\underbrace{u_M,\ldots,u_M}_{n_M}\} \equiv \{(u_1)^{n_1},(u_2)^{n_2},\ldots,(u_M)^{n_M}\}, \qquad (4)$$

where $n_1,\ldots,n_M$ are the multiplicities. Therefore in addition to the BAE, we need to impose extra conditions in order to find physical solutions, similar to the spin-$\frac{1}{2}$ case. The explicit

---

[2]For the BAE of Lieb-Liniger model, this can be proven rigorously. For the XXX spin chain, we have good numerical evidence, but to the best of our knowledge, this has not been proven rigorously.

form of the extra conditions depends on the multiplicities $\{n_1, \ldots, n_M\}$. If we were to derive these extra conditions using ABA, we would have to analyze all possible cases in principle. However, there is strong evidence that such requirements are again encoded automatically in the spin-$s$ $TQ$-relation [27].

The situation for the singular solutions is even more complicated. The spin-$s$ singular solutions contain a Bethe string of length $2s + 1$:

$$\{is, i(s-1), \ldots, -is\}. \tag{5}$$

A generic singular solution can take the following form

$$\{(is)^{m_0}, (i(s-1))^{m_1}, \ldots, (-is)^{m_{2s}}, u_1, u_2, \ldots, u_{M_r}\}, \tag{6}$$

where $\{u_1, \ldots, u_{M_r}\}$ are the regular roots and might contain repeated roots. In principle, one needs to perform an ABA analysis similar to the spin-$\frac{1}{2}$ and derive extra physicality conditions for $\{u_1, \ldots, u_{M_r}\}$. Since we do not have a complete understanding of the possible values of $\{m_0, \ldots, m_{2s}\}$, we have to analyze each case *a priori*, which is a complicated task. The simplest case for $s = 1$ has been investigated in [27], which is already non-trivial. Based on the experience of spin-$\frac{1}{2}$, one can also try to derive these conditions from the rational $Q$-system, if it exists.

The goal of the current work is to construct such a rational $Q$-system for the XXX$_s$ spin chain with periodic and twisted boundary conditions, which generalizes naturally their spin-$\frac{1}{2}$ counterparts. As we shall see, some minor but crucial modifications are required in the higher spin case. The proposed $Q$-system has undergone the following tests: (*i*) For given $L, M, s$, the number of solutions match the ones required by completeness; (*ii*) The numerical Bethe roots obtained from $Q$-system reproduce all the eigenvalues of the Hamiltonian obtained by a direct digonalization; (*iii*) Physicality conditions for some simple singular solutions obtained in [27] can be reproduced from the $Q$-system.

The rest of the paper is structured as follows. In section 2, we present the spin-$s$ rational $Q$-system and numerical tests. In section 3, we first prove that the rational $Q$-system is equivalent to the polynomiality of $TQ$-relation. Based on this, we then prove the completeness of the proposed rational $Q$-system. In section 4, we derive extra conditions for the physical singular solutions. We conclude in section 5 and discuss future directions.

## 2 Rational $Q$-system

In this section, we present the rational $Q$-system for the XXX$_s$ spin chain. We first briefly review some basic facts about spin-$s$ BAE.

### 2.1 BAE and conjecture for completeness

The BAE for spin-$s$ Heisenberg spin chain of length $L$ with $M$ magnons reads

$$\left(\frac{u_j + is}{u_j - is}\right)^L = \prod_{k \neq j}^{M} \frac{u_j - u_k + i}{u_j - u_k - i}, \qquad j = 1, \ldots, M. \tag{7}$$

Here $s$ can be any positive half integer. There are two types of solutions that need special care. One is the solution with *repeated roots* and the other is the so-called *singular solution*. As we emphasize in the introduction, for $s \geq 1$, there are physical solutions that contain repeated roots. They are called *strange solutions*. As for the singular solutions, some of them are physical while others are not. The physical ones satisfy additional constraints, which in principle can be derived using ABA [27].

**Singular solutions**  For singular solutions we have the following important result. Suppose $\{u_k\}$ is a solution of BAE (7) and there exist two Bethe roots, say $u_1, u_2$ whose difference is exactly $i$, namely $u_1 - u_2 = i$, then the solution $\{u_k\}$ must contain an exact Bethe string $\{is, i(s-1), \ldots, -is\}$. The proof of this statement comes from a straightforward but slightly tedious analysis of the zeros and divergences of both sides of (7), which follows closely the spin-$\frac{1}{2}$ case [12] and can be found in the appendix.

For the spin-$s$ BAE, since repeated roots are allowed, a generic singular solution takes the following form

$$\{(is)^{m_0}, (i(s-1))^{m_1}, \ldots, (-is)^{m_{2s}}, u_1, u_2, \ldots, u_{M_r}\}, \tag{8}$$

which means we have $m_k$-fold singular root $i(s-k)$ for $k = 0, 1, \ldots, 2s$ and $M_r$ regular roots. Notice that in the general case, the regular part $\{u_1, \ldots, u_{M_r}\}$ can also contain repeated roots. This means *a priori* we have to analyze all the cases with $m_0, m_1, \ldots, m_{2s} \geq 1$ and derive additional constraints for $\{u_1, \ldots, u_{M_r}\}$. At the moment, we do not have a complete understanding about what are the possible values for $m_0, \ldots, m_{2s}$. What we know for sure is that they cannot be arbitrary. We will derive some constraints for the choices of $\{m_0, \ldots, m_{2s}\}$ in section 4.2. For the special case where $m_0 = m_1 = \ldots = m_{2s} = 1$ and $\{u_1, \ldots, u_{M_r}\}$ all distinct, the extra physical condition derived in [27] is very similar to (2)

$$\prod_{k=1}^{M_r} \left( \frac{u_k + is}{u_k - is} \right)^L = (-1)^L. \tag{9}$$

We will see that this condition can be reproduced from the rational $Q$-system. In addition, we will derive extra conditions for the generic situation (8) in section 4.1.

**Completeness conjecture**  In [27], the authors formulated the following conjecture for the completeness of spin-$s$ Bethe Ansatz

$$\mathcal{N}(L, M) - \mathcal{N}_\text{s}(L, M) + \mathcal{N}_\text{sp}(L, M) + \mathcal{N}_\text{strange}(L, M) = n(L, sL - M), \tag{10}$$

where

$$n(L, r) = b(L, r) - b(L, r + 1), \tag{11}$$

and

$$b(L, r) = \sum_{k=0}^{L} (-1)^k \binom{L}{k} \binom{(s+1)L + r - (2s+1)k - 1}{sL + r - (2s+1)k}. \tag{12}$$

The sum is restricted so that $sL + r - (2s+1)k \geq 0$. On the left-hand side of (10), $\mathcal{N}, \mathcal{N}_\text{s}, \mathcal{N}_\text{sp}$, and $\mathcal{N}_\text{strange}$ denote the number of solutions, singular solutions, singular physical solutions and strange solutions. The authors checked this conjecture numerically for a number of cases.

We have tested extensively that the rational $Q$-system gives exactly $n(L, sL - M)$ solutions for given $L$, $M$, and $s$, which is exactly what is required by the completeness of Bethe Ansatz.

## 2.2  Rational $Q$-system

Let us first introduce two polynomials for later convenience,

$$\alpha(u) = \prod_{k=1}^{2s-1} (u - i(s-k))^L, \qquad q_\alpha(u) = \prod_{k=0}^{2s-1} \left(u - i(s-k-\tfrac{1}{2})\right)^L. \tag{13}$$

In particular, for $s = \frac{1}{2}$, we have $\alpha(u) = 1$ and $q_\alpha(u) = u^L$. To construct the rational $Q$-system, we need the following ingredients:

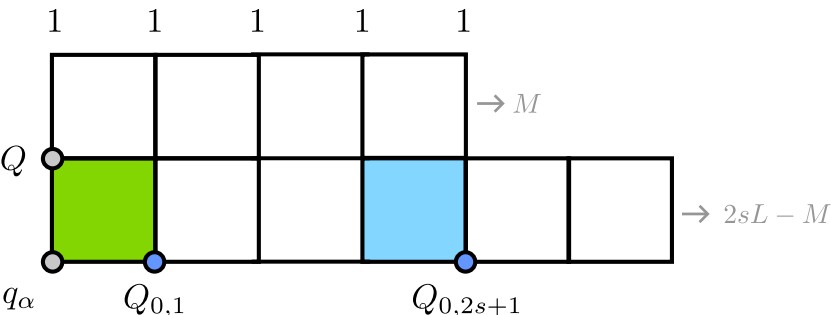

Figure 2: Rational $Q$-system for XXX$_s$ spin chain.

1. A Young tableaux on which we define $Q$-functions;

2. $QQ$-relations which relate the four $Q$-functions defined on each box;

3. Boundary conditions which specify the $Q$-functions partially or completely on the left and upper boundary of the Young tableaux.

**Young tableaux**   The Young tableaux has two rows and is given by $(2sL - M, M)$. On each node, we define a function $Q_{a,b}(u)$, as is shown in Figure 2.

**The QQ-relation**   The $QQ$-relation is the same as the spin-$\frac{1}{2}$ case,[3]

$$Q_{a,b+1}Q_{a+1,b} = Q_{a,b}^{+}Q_{a+1,b+1}^{-} - Q_{a,b}^{-}Q_{a+1,b+1}^{+}. \tag{14}$$

Here and in what follows, we have introduced the standard shorthand notation $F^{\pm}(u) \equiv F(u \pm \frac{i}{2})$.

**Boundary conditions**   The $Q$-functions at the upper and left boundary of the Young tableaux are given by

$$Q_{2,b} = 1, \qquad b = 0, 1, \ldots, M, \tag{15}$$
$$Q_{1,0} = Q(u),$$
$$Q_{0,0} = q_\alpha(u),$$

where $q_\alpha(u)$ has been defined in (13). The zeros of $Q(u)$ are the Bethe roots that we want to find. In the generic situation, we can write

$$Q(u) = f(u)\mathcal{Q}(u), \qquad f(u) = \prod_{k=0}^{2s}(u - i(s-k))^{m_k}, \qquad \mathcal{Q}(u) = \prod_{j=1}^{M_r}(u - u_j), \tag{16}$$

such that the zeros of $f(u)$ and $\mathcal{Q}(u)$ contain singular and regular roots respectively. We have

$$M_r + m_0 + m_1 + \ldots + m_{2s} = M. \tag{17}$$

---

[3]Notice that the $QQ$-relation is defined up to proportionality, so multiplying any constant on the right-hand side defines the same $Q$-system.

### 2.3 Twisted boundary condition

It is straightforward to generalize the spin-$s$ rational $Q$-system to a slightly different case, the XXX$_s$ chains with the twisted boundary condition. For a length-$L$ spin chain, instead of the periodic boundary condition, we impose

$$S^z_{L+1} = S^z_1, \quad S^\pm_{L+1} = e^{\pm i\theta}S^\pm_1, \tag{18}$$

with a twisting angle $\theta$. Here $S^\alpha_n$ ($\alpha = \pm, z$) is the local spin-$s$ generator of SU(2) symmetry acting on site-$n$. In the untwisted limit $\theta \to 0$, one recovers the periodic spin chain. The twisted boundary condition breaks the SU(2) symmetry of the closed spin chain into U(1), which preserves the number of magnons. For the spin-$\frac{1}{2}$ case, it is known that one can equivalently consider the following Hamiltonian for the twisted spin chain, (see *e.g.* [21])

$$\mathcal{H}_\theta = 4\sum_{j=1}^{L}\left(\frac{1}{4} - S^3_j S^3_{j+1} - \frac{e^{i\theta/L}}{2}S^+_j S^-_{j+1} - \frac{e^{i\theta/L}}{2}S^-_j S^+_{j+1}\right), \tag{19}$$

which describes a circular spin chain with a magnetic flux passing through. The XXX$_s$ spin chain with twisted boundary condition (18) (corresponding to the so-called diagonal twist) still preserves U(1) symmetry and can be solved by algebraic Bethe Ansatz, leading to the slightly modified BAE

$$e^{i\theta}\left(\frac{u_i + is}{u_i - is}\right)^L \prod_{j=1,j\neq i}^{M}\frac{u_i - u_j - i}{u_i - u_j + i} = 1. \tag{20}$$

One can check readily that the above BAE can be derived from a slightly modified $Q$-system where the $QQ$-relation (14) becomes

$$Q_{a,b+1}Q_{a+1,b} = Q^+_{a,b}Q^-_{a+1,b+1} - \kappa_a Q^-_{a,b}Q^+_{a+1,b+1}, \tag{21}$$

with $\kappa_1 = 1$ and $\kappa_0 = e^{i\theta}$, encoding the twisted angle. The boundary conditions remain the same as (15).

By solving the above proposed rational $Q$-system for the twisted case, we have checked extensively that the $Q$-system indeed gives complete and only physical solutions of the twisted BAE (20), more details can be found in section 2.5. However, proving this fact rigorously like in the periodic case is beyond the scope of the current work. Indeed, one of the main results that we use for the proof of completeness in the periodic boundary condition is that both solutions of the $TQ$-relation are polynomials. This is no longer the case for the twisted boundary condition, see for example [28] for the spin-1/2 case and [29] for the higher spin case. Therefore it is non-trivial to generalize the proof in this paper to the twisted case and we prefer to present a more detailed discussion elsewhere [30].

### 2.4 Solving rational $Q$-system

For the spin-$\frac{1}{2}$ case, in order to solve the rational $Q$-system, we first make an Ansatz for $Q(u)$ and then require all the $Q$-functions on the Young tableaux to be polynomials. For the spin-$s$ case, this has to be modified slightly.

For the BAE with $M$ magnons, we make the usual Ansatz

$$Q(u) = u^M + \sum_{k=0}^{M-1} c_k u^k. \tag{22}$$

We then require $Q_{0,1}/\alpha(u)$ and $Q_{0,b}$ ($b = 2, 3, \ldots, 2sL - M$) to be polynomials. Notice that the first condition is special. In the spin-$\frac{1}{2}$ case, $\alpha(u)$ defined in (13) is trivial, so we do

not see this difference. At the moment it might seem a bit strange that the first $Q_{0,1}$ is treated somewhat differently. However, we will prove that the condition $Q_{0,1}/\alpha(u)$ being a polynomial is equivalent to the $TQ$-relation.

**$TQ$-relation**   For XXX$_s$ chain, Baxter's $TQ$-relation reads

$$\tau(u)Q(u) = (u+is)^L Q(u-i) + (u-is)^L Q(u+i)\,, \tag{23}$$

where $\tau(u)$ is the eigenvalue of the transfer matrix, which is a polynomial of degree $L$. Taking $u = u_k$ where $u_k$ are the zeros of $Q(u)$, we recover the BAE. The $TQ$-relation can be viewed as a second-order difference equation for the unknown function $Q(u)$. In general, there are two linearly independent solutions. Therefore for each polynomial solution $Q(u)$ with degree $M \le [sL]$, there exists a dual solution of the $TQ$-relation, which we denote by $P(u)$. As usual, $P(u)$ and $Q(u)$ satisfy the Wronskian relation

$$P^+ Q^- - P^- Q^+ = q_\alpha(u)\,. \tag{24}$$

We shall prove this result in the next section.

**Solution of $QQ$-relation**   Using the dual solution $P(u)$, we can write down the solution of the $QQ$-relation

$$
\begin{aligned}
Q_{1,n} &= D^n Q\,, \quad n = 0,\dots,M\,, \\
Q_{0,n} &= D^n P^+ D^n Q^- - D^n P^- D^n Q^+\,, \quad n = 0,\dots,2sL - M\,,
\end{aligned}
\tag{25}
$$

where we have introduced the operator $D$ such that

$$Df(u) \equiv f\left(u + \tfrac{i}{2}\right) - f\left(u - \tfrac{i}{2}\right)\,. \tag{26}$$

The solution of $Q_{1,n}$ is obvious. The solution of $Q_{0,n}$ can be proved by mathematical induction. Notice that for $n = 0$, it is nothing but the Wronskian relation (24).

**A sequence of $TQ$-relations**   In what follows, it is useful to introduce a sequence of $T$-functions, defined as

$$T_n = Q_{0,n}^+ + Q_{0,n}^- - Q_{0,n+1}\,, \qquad n = 0, 1, \dots \tag{27}$$

In particular, we have

$$T_0(u) = \tau(u)\alpha(u)\,. \tag{28}$$

This can be proven as follows. Multiplying both sides of the $TQ$-relation by $\alpha(u)$, we obtain

$$\tau(u)\alpha(u)Q(u) = q_\alpha^+ Q^{--} + q_\alpha^- Q^{++}\,. \tag{29}$$

Plugging the Wronskian relation (24) in the right-hand side of the above equation leads to

$$\tau(u)\alpha(u) = P^{++} Q^{--} - P^{--} Q^{++}\,. \tag{30}$$

On the other hand, taking $n = 0$ in (27), we have

$$T_0 = Q_{0,0}^+ + Q_{0,0}^- - Q_{0,1}\,. \tag{31}$$

Plugging in the solution of the *QQ*-relation (25), we find

$$T_0 = P^{++}Q^{--} - P^{--}Q^{++}.$$ (32)

Comparing (30) and (32), we have (28). Using $T_n$, the rational $Q$-system can be written in terms of a sequence of *TQ*-like relations

$$T_n Q_{1,n} = Q_{0,n}^+ Q_{1,n}^{--} + Q_{0,n}^- Q_{1,n}^{++}, \quad n = 0, \ldots, 2sL - M.$$ (33)

$T_n$ can also be expressed compactly in terms of $P(u)$ and $Q(u)$

$$T_n = (D^n P)^{++}(D^n Q)^{--} - (D^n P)^{--}(D^n Q)^{++}.$$ (34)

This can be proven straightforwardly by mathematical induction from the definition of $T_n$ and the solution of $Q_{0,n}$.

**Why is $Q_{0,1}$ special** Now we can explain why we divide $Q_{0,1}$ by $\alpha(u)$. We have

$$T_0(u) = \tau(u)\alpha(u) = Q_{0,0}^+ + Q_{0,0}^- - Q_{0,1}.$$ (35)

Notice that *TQ*-relation requires that $\tau(u)$ instead of $T_0(u)$ to be a polynomial. We have

$$\tau(u) = \frac{Q_{0,0}^+}{\alpha} + \frac{Q_{0,0}^-}{\alpha} - \frac{Q_{0,1}}{\alpha} = \frac{q_\alpha^+}{\alpha} + \frac{q_\alpha^-}{\alpha} - \frac{Q_{0,1}}{\alpha}$$
$$= (u + is)^L + (u - is)^L - \frac{Q_{0,1}}{\alpha},$$ (36)

which implies that $\tau(u)$ is a polynomial if and only if $Q_{0,1}/\alpha$ is a polynomial. Therefore we see that the *QQ*-relations on the box colored green in Figure 2, together with the requirement that $Q_{0,1}/\alpha$ being a polynomial is equivalent to the *TQ*-relation.

## 2.5 Numerical checks

In order to test the correctness of the proposed $Q$-system, we have performed extensive numerical checks for various $L, M, s$. There are two direct tests we can perform: one is to count the number of physical solutions, *i.e.* testing the completeness of the results obtained from the proposed $Q$-system; the second one is to calculate the energy spectrum of the corresponding XXX$_s$ spin chain and compare with the results obtained from a brute-force diagonalization of the Hamiltonian.

**Number of solutions** The expected number of physical solutions for a spin-$s$ $L$-site chain with $M$ magnons is given by [31, 32]

$$\mathcal{N}_s(L, M) = c_s(L, M) - c_s(L, M - 1),$$ (37)

with

$$c_s(L; M) = \sum_{j=0}^{\lfloor \frac{L+M-1}{2s+1} \rfloor} (-1)^j \binom{L}{j} \binom{L + M - 1 - (2s + 1)j}{L - 1}.$$ (38)

Note that in (37) we only consider $M \leq sL$ in order to count the number of the highest-weight states. The number of SU(2) descendant states can be taken into account straightforwardly from representation theory. We checked the matching of the number of solutions in spin-1 chains up to $L = 6$ sites, spin-$\frac{3}{2}$ chains up to $L = 5$ and spin-2 chains up to $L = 4$ for all $M \leq sL$. For $M = 1, 2, 3$, we checked the matching in spin-1 chains up to $L = 12$ sites, spin-$\frac{3}{2}$ chains up to $L = 10$, and spin-2 chains up to $L = 8$. Some more precise information about the solutions that are difficult to access in traditional approaches is listed in Table 1 and 2.

Table 1: Number of physical singular solutions and repeated singular solutions in spin-1 chains. $\mathcal{N}_{\text{phys}}$, $\mathcal{N}_{\text{sing}}$ and $\mathcal{N}_{\text{rep. sing}}$ respectively denote the number of all physical solutions, physical singular solutions and repeated singular solutions obtained from the $Q$-system.

|  | $\mathcal{N}_{\text{phys}}$ | $\mathcal{N}_{\text{sing}}$ | $\mathcal{N}_{\text{rep. sing}}$ |
|---|---|---|---|
| $L = 3, M = 1, s = 1$ | 2 | 0 | 0 |
| $L = 3, M = 2, s = 1$ | 3 | 0 | 0 |
| $L = 3, M = 3, s = 1$ | 1 | 1 | 0 |
| $L = 6, M = 4, s = 1$ | 40 | 1 | 1 |
| $L = 6, M = 5, s = 1$ | 36 | 3 | 0 |
| $L = 6, M = 6, s = 1$ | 15 | 2 | 2 |
| $L = 9, M = 1, s = 1$ | 8 | 0 | 0 |
| $L = 9, M = 2, s = 1$ | 36 | 0 | 0 |
| $L = 9, M = 3, s = 1$ | 111 | 1 | 0 |
| $L = 9, M = 4, s = 1$ | 258 | 0 | 0 |
| $L = 12, M = 3, s = 1$ | 274 | 1 | 0 |
| $L = 12, M = 4, s = 1$ | 869 | 1 | 1 |

Table 2: Number of physical singular solutions and repeated singular solutions in spin-$\frac{3}{2}$ and spin-2 chains. $\mathcal{N}_{\text{phys}}$, $\mathcal{N}_{\text{sing}}$ and $\mathcal{N}_{\text{rep. sing}}$ respectively denote the number of all physical solutions, physical singular solutions and repeated singular solutions obtained from the $Q$-system. We notice that in [27] it was suspected that there is a repeated singular solution in the case of $L = 8, M = 6, s = \frac{3}{2}$, but in our approach, it is easy to see that all the singular solutions have no repeated Bethe roots.

|  | $\mathcal{N}_{\text{phys}}$ | $\mathcal{N}_{\text{sing}}$ | $\mathcal{N}_{\text{rep. sing}}$ |
|---|---|---|---|
| $L = 8, M = 4, s = \frac{3}{2}$ | 202 | 1 | 0 |
| $L = 8, M = 6, s = \frac{3}{2}$ | 700 | 4 | 0 |
| $L = 9, M = 4, s = \frac{3}{2}$ | 321 | 0 | 0 |
| $L = 10, M = 4, s = \frac{3}{2}$ | 485 | 1 | 0 |
| $L = 6, M = 5, s = 2$ | 120 | 1 | 0 |
| $L = 6, M = 6, s = 2$ | 180 | 1 | 1 |
| $L = 7, M = 5, s = 2$ | 245 | 1 | 0 |
| $L = 7, M = 6, s = 2$ | 420 | 0 | 0 |

**Physical spectrum** In the brute-force diagonalization approach, we directly compute the eigenvalues of the XXX$_s$ Hamiltonian [1, 33]

$$\mathcal{H} = \sum_{j=1}^{N} Q_{2s}(\vec{S}_j \cdot \vec{S}_{j+1}),$$
(39)

where

$$Q_{2s}(x) := \sum_{j=1}^{2s} h(j) \prod_{\substack{l=0 \\ l \neq j}}^{2s} \frac{x - x_l}{x_j - x_l}, \quad x_l = \frac{1}{2} l(l+1) - s(s+1),$$
(40)

and $h(j)$ is the $j$-th harmonic number given by

$$h(j) = \sum_{k=1}^{j} \frac{1}{k}.$$
(41)

To compare with the spectrum computed from the Bethe Ansatz equation [1, 33, 34]

$$E_s = \sum_{i=1}^{M} \frac{s}{u_i^2 + s^2},$$
(42)

where $\{u_i\}_{i=1}^{M}$ is the set of Bethe roots, we normalize the Hamiltonian such that the pseudovacuum has zero energy. For $s = 1, \frac{3}{2}, 2$, the explicit expressions of the Hamiltonian are

$$\mathcal{H}_1 = -\frac{1}{4} \sum_{j=1}^{L} \left( \vec{S}_j^{(1)} \cdot \vec{S}_{j+1}^{(1)} + (\vec{S}_j^{(1)} \cdot \vec{S}_{j+1}^{(1)})^2 \right),$$
(43)

$$\mathcal{H}_{\frac{3}{2}} = \frac{1}{432} \sum_{j=1}^{L} \left( 27 \vec{S}_j^{(\frac{3}{2})} \cdot \vec{S}_{j+1}^{(\frac{3}{2})} - 8(\vec{S}_j^{(\frac{3}{2})} \cdot \vec{S}_{j+1}^{(\frac{3}{2})})^2 - 16(\vec{S}_j^{(\frac{3}{2})} \cdot \vec{S}_{j+1}^{(\frac{3}{2})})^3 \right) + \frac{3L}{8} \text{Id}_{4^L \times 4^L},$$
(44)

$$\mathcal{H}_2 = -\frac{1}{864} \sum_{j=1}^{L} \left( 234 \vec{S}_j^{(2)} \cdot \vec{S}_{j+1}^{(2)} + 43(\vec{S}_j^{(2)} \cdot \vec{S}_{j+1}^{(2)})^2 - 10(\vec{S}_j^{(2)} \cdot \vec{S}_{j+1}^{(2)})^3 - 3(\vec{S}_j^{(2)} \cdot \vec{S}_{j+1}^{(2)})^4 \right)$$
$$+ \frac{L}{4} \text{Id}_{5^L \times 5^L},$$
(45)

with $\vec{S}^{(s)} = \frac{1}{2}(X^{(s)}, Y^{(s)}, Z^{(s)})$. By our convention, the matrices read

$$X^{(1)} = \sqrt{2} \begin{pmatrix} 0 & 1 & 0 \\ 1 & 0 & 1 \\ 0 & 1 & 0 \end{pmatrix}, \quad Y^{(1)} = \sqrt{2} \begin{pmatrix} 0 & -i & 0 \\ i & 0 & -i \\ 0 & i & 0 \end{pmatrix}, \quad Z^{(1)} = \begin{pmatrix} 2 & 0 & 0 \\ 0 & 0 & 0 \\ 0 & 0 & -2 \end{pmatrix}, \quad (46)$$

$$X^{(\frac{3}{2})} = \begin{pmatrix} 0 & \sqrt{3} & 0 & 0 \\ \sqrt{3} & 0 & 2 & 0 \\ 0 & 2 & 0 & \sqrt{3} \\ 0 & 0 & \sqrt{3} & 0 \end{pmatrix}, \quad Y^{(\frac{3}{2})} = \begin{pmatrix} 0 & -i\sqrt{3} & 0 & 0 \\ i\sqrt{3} & 0 & -2i & 0 \\ 0 & 2i & 0 & -i\sqrt{3} \\ 0 & 0 & i\sqrt{3} & 0 \end{pmatrix}, \quad (47)$$

$$Z^{(\frac{3}{2})} = \begin{pmatrix} 3 & 0 & 0 & 0 \\ 0 & 1 & 0 & 0 \\ 0 & 0 & -1 & 0 \\ 0 & 0 & 0 & -3 \end{pmatrix},$$

and

$$X^{(2)} = \begin{pmatrix} 0 & 2 & 0 & 0 & 0 \\ 2 & 0 & \sqrt{6} & 0 & 0 \\ 0 & \sqrt{6} & 0 & \sqrt{6} & 0 \\ 0 & 0 & \sqrt{6} & 0 & 2 \\ 0 & 0 & 0 & 2 & 0 \end{pmatrix}, \quad Y^{(2)} = \begin{pmatrix} 0 & -2i & 0 & 0 & 0 \\ 2i & 0 & -i\sqrt{6} & 0 & 0 \\ 0 & i\sqrt{6} & 0 & -i\sqrt{6} & 0 \\ 0 & 0 & i\sqrt{6} & 0 & -2i \\ 0 & 0 & 0 & 2i & 0 \end{pmatrix}, \quad (48)$$

$$Z^{(2)} = \begin{pmatrix} 4 & 0 & 0 & 0 & 0 \\ 0 & 2 & 0 & 0 & 0 \\ 0 & 0 & 0 & 0 & 0 \\ 0 & 0 & 0 & -2 & 0 \\ 0 & 0 & 0 & 0 & -4 \end{pmatrix}.$$

To simplify the computation in the $Q$-system approach, it is more convenient to rewrite the expression of the energy (42) in terms of the coefficients $c_k$ of the $Q$-function defined in (22). For example, for $M = 2$, we have

$$E_s(c_0, c_1) = \frac{2s^3 - 2sc_0 + sc_1^2}{s^4 - 2s^2c_0 + s^2c_1^2 + c_0^2}. \quad (49)$$

The calculation becomes tricky for physical singular solutions. Apparent divergence appears in the evaluation of the energy when two Bethe roots take the value $\{u_1 = is, u_2 = -is\}$, but the energy can be regularized to a finite value if we set $u_1 = is + \epsilon$, $u_2 = -is + \epsilon$, and by keeping the $\epsilon^0$-order, we obtain a regularized expression of the energy in terms of the remaining $M-2$ Bethe roots,

$$\bar{E}_s = \sum_{i=3}^{M} \frac{s}{u_i^2 + s^2} + \frac{1}{2s}. \quad (50)$$

One can further define the elementary symmetric polynomial of the remaining $M-2$ Bethe roots as

$$d_k = e_{M-2-k}(\{-u_i\}_{i=3}^{M}), \quad (51)$$

then it is easy to derive the relation between $d_k$ and $c_k$,

$$c_{M-k} = d_{M-2-k} + s^2 d_{M-k}, \quad (52)$$

and the regularized energy is given by

$$\bar{E}_s(\{c_k\}_{k=0}^{M-1}) = E_s(\{d_k\}_{k=0}^{M-3}) + \frac{1}{2s}. \quad (53)$$

We checked that in the case of spin-1 chains for $L = 2, 3, 4, 5$, in spin-$\frac{3}{2}$ chains for $L = 2, 3, 4$ and in spin-2 chains for $L = 2, 3$, the energy spectra match perfectly between the brute-force diagonalization and the Bethe Ansatz $Q$-system. As an example, we list the energy spectrum of $L = 3$, $s = 1$ chain: $\{0, \frac{3}{4}, \frac{3}{2}, \frac{7}{4}, \frac{5}{2}\}$.

**Twisted boundary condition** We also performed numerical checks for the $Q$-system of twisted XXX$_s$ spin chain, (21). When the twisted boundary condition (18) is imposed, the symmetry is broken from SU(2) to U(1), and the solutions to the BAE are no longer the highest-weight states of SU(2). The solutions to the BAE in the twisted case thus recombine into SU(2) highset-weight multiplets in the untwisted limit $\theta = 0$ following the rule of the representation theory, (37). We verified the number of solutions given by the $Q$-system agrees with the expected number $c_s(L, M)$ from the representation theory, for $s = 1$ $M = 1, 2, 3, 4$ up to $L = 12$,

and $s = 2$ $M = 5, 6$ up to $L = 7$, and for $s = \frac{5}{2}$ we checked the matching for $M = 3, 4$ up to $L = 9$.

We have some interesting observations from our numerical results. For the $Q$-system with a generic twisted angle, namely $\theta \neq 0, \pi$, it seems that there are no physical singular solutions or strange solutions (*i.e.* solutions containing the exact string $\{-is, -i(s-1), \ldots, is\}$). This is in contrast to the untwisted case (for spin-$\frac{1}{2}$, it has been conjectured in [21]). So far we do not have a proof for this fact. If this were the case, the analysis of XXX$_s$ spin chains with twisted boundary will be largely simplified, and the completeness problem of the periodic spin chains of arbitrary spin can thus be accessed from the analysis of the twisted case. More numerical results and discussions will be presented in [30].

## 3   *TQ*-relation and polynomiality

In this section, we prove the following results rigorously

1. The Wronskian relation (24);

2. $P(u)$ is a polynomial if and only if $Q_{0,1}/\alpha$ and $Q_{0,2s+1}$ are polynomials.

We shall formulate the two results as two theorems. From the second result, we can infer the minimality condition for the rational $Q$-system. Namely, if $Q_{0,1}/\alpha$ and $Q_{0,2s+1}$ are polynomials it follows that the rest of the $Q_{0,n}$ are also polynomials. This shows that in the spin-$s$ case, the rational $Q$-system we proposed is also equivalent to imposing polynomiality of $\tau(u)$, $Q(u)$, and $P(u)$.

### 3.1   Dual solution and Wronskian relation

**Thoerem 1**   (**Wronskian relation**) Consider *TQ*-relation

$$\tau(u)Q(u) = (u+is)^L Q(u-i) + (u-is)^L Q(u+i), \tag{54}$$

where $\tau(u)$ is a polynomial. For each polynomial solution $Q(u)$ with $\deg Q \leq [sL]$, one can construct a function $P(u)$ which satisfies the following Wronskian relation,

$$P^+ Q^- - P^- Q^+ = q_\alpha. \tag{55}$$

As a result of (54) and (55), $P(u)$ is the other solution (the dual solution of $Q(u)$) of the *TQ*-relation with the same $\tau(u)$.

*Proof* The polynomial $Q(u)$ in general has the following structure

$$Q(u) = \prod_{k=0}^{2s} (u - i(s-k))^{m_k} \times \mathcal{Q}(u) \equiv f(u)\mathcal{Q}(u), \tag{56}$$

where we have separated the singular and regular roots. Let us denote $\deg \mathcal{Q} = M_r$, we have

$$\deg Q = \deg \mathcal{Q} + \deg f = M_r + \sum_{k=0}^{2s} m_k. \tag{57}$$

We start with the *TQ*-relation (multiplied by $\alpha(u)$ on both sides)

$$\tau \alpha Q = q_\alpha^+ Q^{--} + q_\alpha^- Q^{++}. \tag{58}$$

Dividing both sides by $Q^{--}QQ^{++}$, we obtain

$$\frac{\tau\alpha}{Q^{--}Q^{++}} = R^+ + R^-, \qquad R \equiv \frac{q_\alpha}{Q^-Q^+}. \tag{59}$$

We perform the partial fraction expansion for $R(u)$

$$R(u) = \frac{q_\alpha}{f^+f^-Q^+Q^-} = \pi(u) + \frac{q_+}{Q^+} + \frac{q_-}{Q^-} + \frac{q_s}{f^+f^-}, \tag{60}$$

where $\pi(u), q_\pm(u)$ and $q_s(u)$ are polynomials. Plugging back to (59), we have

$$\frac{\tau\alpha}{Q^{--}Q^{++}} = \pi^+ + \pi^- + \frac{q_+^+}{Q^{++}} + \frac{q_-^-}{Q^{--}} + \frac{q_+^- + q_-^+}{Q} + \frac{q_s^+}{f^{++}f} + \frac{q_s^-}{ff^{--}}. \tag{61}$$

Let us first consider the non-singular roots $\{u_k\}$. Suppose $u_k$ has multiplicity $n_k$. The left-hand side of (61) is regular at $u = u_k$. As a result, the right-hand side should also be regular at $u = u_k$. The term $(q_+^- + q_-^+)/Q$ seems to have an $n_k$-th order pole at $u = u_k$. We thus conclude that this pole must be spurious, which implies the following $n_k$ constraints,

$$\frac{\partial^n}{\partial u^n}\left[q_+\left(u - \tfrac{i}{2}\right) + q_-\left(u + \tfrac{i}{2}\right)\right]\Big|_{u=u_k} = 0, \qquad n = 0, 1, \ldots, n_k - 1, \tag{62}$$

at $u = u_k$. Similarly, at all regular roots we have constraints like (62). Since $\deg(q_\pm) < M_r(= \deg(Q))$, it is easy to see that the only way to fulfill all the constraints is that this term vanish, *i.e.* $q_+(u) + q_-(u) = 0$. We can therefore write the two polynomials $q_+(u)$ and $q_-(u)$ in terms of only one polynomial $q(u)$ as follows,

$$q_+(u) = q\left(u + \tfrac{i}{2}\right), \qquad q_-(u) = -q\left(u - \tfrac{i}{2}\right). \tag{63}$$

After this analysis, we find that

$$\frac{\tau\alpha}{Q^{--}Q^{++}} = \pi^+ + \pi^- + \frac{q^{++}}{Q^{++}} - \frac{q^{--}}{Q^{--}} + \frac{q_s^+}{f^{++}f} + \frac{q_s^-}{ff^{--}}. \tag{64}$$

Now we analyze the part that contains exact strings. Let us define

$$U(u) \equiv \frac{q_s}{f^+f^-} = q_s \prod_{k=0}^{2s} \frac{1}{\left(u - i(s-k+\tfrac{1}{2})\right)^{m_k}\left(u - i(s-k+\tfrac{1}{2})\right)^{m_k}} \tag{65}$$

$$= \frac{q_s}{\left(u - i(s+\tfrac{1}{2})\right)^{m_0}\left(u + i(s+\tfrac{1}{2})\right)^{m_{2s}}} \prod_{k=1}^{2s} \frac{1}{\left(u - i(s-k+\tfrac{1}{2})\right)^{n_k}},$$

where $n_k = m_{k-1} + m_k$ for $k = 1, \ldots, 2s$. We now perform the partial fraction decomposition for $U(u)$

$$U(u) = \sum_{k=1}^{2s}\sum_{m=1}^{n_k} \frac{b_k^{(m)}}{\left(-iu - s + k - \tfrac{1}{2}\right)^m} + \sum_{m=1}^{m_0} \frac{b_0^{(m)}}{\left(-iu - s - \tfrac{1}{2}\right)^m} + \sum_{m=1}^{m_{2s}} \frac{b_{2s}^{(m)}}{\left(-iu + s + \tfrac{1}{2}\right)^m}. \tag{66}$$

Our goal is to write $U(u)$ in a difference form $U(u) = V^+(u) - V^-(u)$ for certain properly defined $V(u)$. For the last two terms in (66), we can add and subtract terms to bring them in difference forms. The new terms we introduced in the process can be combined with the first sum, leading to some shifts on the parameters $b_k^{(m)}$. More explicitly, we have

$$U(u) = \sum_{k=1}^{2s}\sum_{m=1}^{n_k} \frac{\tilde{b}_k^{(m)}}{\left(-iu - s + k - \tfrac{1}{2}\right)^m} - \sum_{m=1}^{m_0}\left(\frac{b_0^{(m)}}{\left(-iu - s + \tfrac{1}{2}\right)^m} - \frac{b_0^{(m)}}{\left(-iu - s - \tfrac{1}{2}\right)^m}\right) \tag{67}$$

$$+ \sum_{m=1}^{m_{2s}}\left(\frac{b_{2s}^{(m)}}{\left(-iu + s + \tfrac{1}{2}\right)^m} - \frac{b_{2s}^{(m)}}{\left(-iu + s - \tfrac{1}{2}\right)^m}\right),$$

where the last two sums are in the difference form and the coefficients $\tilde{b}_k^{(m)}$ in the first line have been shifted.[4] To write the first sum in a difference form, we introduce the polygamma functions

$$\psi^{(n)}(z) = \frac{\mathrm{d}^{n+1}}{\mathrm{d}z^{n+1}} \ln \Gamma(z), \quad n = 0, 1, \dots \tag{68}$$

We then have

$$\frac{1}{\left(-iu - s + k - \frac{1}{2}\right)^m} = v_k^{(m)}\left(u + \frac{i}{2}\right) - v_k^{(m)}\left(u - \frac{i}{2}\right), \tag{69}$$

where

$$v_k^{(m)}(u) = \frac{(-1)^m}{(m-1)!} \psi^{(m-1)}(-iu - s + k). \tag{70}$$

Therefore we can write

$$U(u) = V\left(u + \frac{i}{2}\right) - V\left(u - \frac{i}{2}\right), \tag{71}$$

with

$$V(u) = \sum_{k=1}^{2s} \sum_{m=1}^{n_k} \tilde{b}_k^{(m)} v_k^{(m)}(u) - \sum_{m=1}^{m_0} \frac{b_0^{(m)}}{(-iu+s)^m} + \sum_{m=1}^{m_{2s}} \frac{b_{2s}^{(m)}}{(-iu+s)^m}. \tag{72}$$

Combining (63) and (71), we have

$$\frac{q_\alpha}{Q^+ Q^-} = \pi(u) + \frac{q^+}{Q^+} - \frac{q^-}{Q^-} + U(u) \tag{73}$$

$$= \rho^+ - \rho^- + \frac{q^+}{Q^+} - \frac{q^-}{Q^-} + V^+ - V^-,$$

where we have introduced another polynomial $\rho(u)$ to write $\pi(u)$ in a difference form. Such a polynomial always exist and is defined up to a constant. We can write the right-hand side of (73) in the form

$$\frac{q_\alpha}{Q^+ Q^-} = \frac{P^+}{Q^+} - \frac{P^-}{Q^-} = \frac{P^+ Q^- - P^- Q^+}{Q^+ Q^-}, \tag{74}$$

where

$$P(u) = [q(u)f(u) + Q(u)\rho(u)] + Q(u)V(u) = P_0(u) + Q(u)V(u). \tag{75}$$

Comparing the numerator of both sides of (74), we have

$$P^+ Q^- - P^- Q^+ = q_\alpha. \tag{76}$$

Now we show that $P(u)$ is the solution of $TQ$-relation. Multiplying both sides of (54) by $\alpha(u)$ we obtain

$$\tau(u)\alpha(u)Q(u) = q_\alpha^+ Q^{--} + q_\alpha^- Q^{++}. \tag{77}$$

Plugging in the Wronskian relation (55), we obtain

$$\tau(u)\alpha(u) = P^{++} Q^{--} - P^{--} Q^{++}. \tag{78}$$

Now multiply both sides of (78) by $P(u)$, we have

$$\tau(u)\alpha(u)P(u) = (P^{++}Q^{--} - P^{--}Q^{++})P \tag{79}$$

$$= P^{++}\left(Q^{--}P - QP^{--}\right) + P^{--}\left(P^{++}Q - PQ^{++}\right)$$

$$= q_\alpha^+ P^{--} + q_\alpha^- P^{++}.$$

---

[4]In fact, only $b_1^{(m)}$ and $b_{2s}^{(m)}$ are shifted, the rest coefficients are not modified. But this is not relevant for our proof.

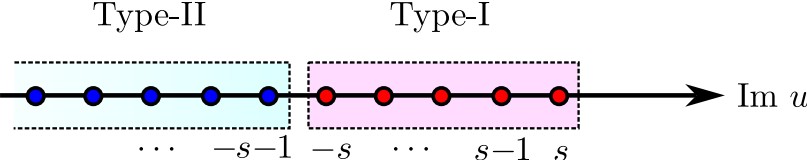

Figure 3: Potential poles of $P(u)$. The red bullets denote Type-I poles and the blue bullets denote Type-II poles.

Dividing both sides of the above relation by $\alpha(u)$ leads to

$$\tau(u)P(u) = (u+is)^L P^{--} + (u-is)^L P^{++}. \tag{80}$$

This proves that $P(u)$ is the dual solution of the $TQ$-relation.

A few remarks are in order. First of all, the dual solution $P(u)$ is not unique. Any shift of the form $P(u) \to P(u) + \beta Q(u)$ leaves the Wronskian relation invariant. Secondly, due to the presence of the term $Q(u)V(u)$ which contains polygamma functions, $P(u)$ is in general not a polynomial.

### 3.2 Polynomiality condition for $P(u)$

From the construction of $P(u)$ in the previous subsection, we see that it is not a polynomial in general. In this subsection, we give the condition under which $P(u)$ becomes a polynomial. We have the following key result.

**Theorem 2** (**Polynomiality condition**) $P(u)$ is a polynomial if $T_0(u)/\alpha(u)$ and $T_{2s}(u)$ are polynomials, where $T_n$ has been defined in (27).

Our strategy for the proof is as follows. Since $P(u)$ is the sum of polynomials and polygamma functions, its analytic property is rather simple. The only singularities are poles on the complex $u$-plane. We will show that if $T_0/\alpha$ and $T_{2s}$ are polynomials, $P(u)$ becomes regular at all these potential poles, which implies that $P(u)$ is a polynomial.

From the explicit form of $P(u)$ (75), we can write down the potential poles. It is convenient to classify them into two types, which we shall call type-I and type-II. They are given by

$$\begin{aligned}
\text{Type-I}: \quad & u = is, is-i, \ldots, -is, \\
\text{Type-II}: \quad & u = -is-ni, \qquad n = 1, 2, \ldots,
\end{aligned} \tag{81}$$

as is shown in Figure 3. We have the following two lemmas concerning these two types of poles.

**Lemma 1** $P(u)$ is regular at Type-I poles if $T_0/\alpha$ is a polynomial.
*Proof.* We have the following relations

$$\begin{aligned}
T_0(u) &= \tau(u)\alpha(u) = P^{++}Q^{--} - P^{--}Q^{++}, \\
q_\alpha(u) &= P^+Q^- - P^-Q^+.
\end{aligned} \tag{82}$$

Eliminating $Q(u)$ from the previous two equations, we obtain

$$q_\alpha^+ P^{--} + q_\alpha^- P^{++} = P\, T_0, \tag{83}$$

which can be written as

$$P^{--}(u) = P(u)\frac{\tau(u)}{(u+is)^L} - P^{++}(u)\left(\frac{u-is}{u+is}\right)^L, \tag{84}$$

where we have used $T_0(u) = \alpha(u)\tau(u)$. Taking $u = is$ in (84), we obtain

$$P(i(s-1)) = P(is)\frac{\tau(is)}{(2is)^L}. \tag{85}$$

By our assumption $T_0/\alpha = \tau(u)$ is a polynomial, so it is regular on the whole complex plane. Notice that $P(u)$ is in fact regular at $u = is$ because the pole in $V(u)$ cancelled neatly with the zero in $Q(u)$ so that $Q(u)V(u)$ in (75) is regular at $u = is$. It then follows from (85) that $P(i(s-1))$ is finite. From (84), taking $u = i(s-k)$, we have the following recursion relation,

$$P(i(s-k)-i) = P(i(s-k))\frac{\tau(i(s-k))}{(2is-ik)^L} - P(i(s-k)+i)\left(\frac{k}{k-2s}\right)^L. \tag{86}$$

This can be used to prove recursively that $P(i(s-k)-i)$'s are finite for $k = 1, 2, \ldots, 2s-1$. Therefore $P(u)$ is regular at $u = is, is-1, \ldots, -is$, which completes the proof.

In fact from the assumption that $T_0/\alpha$ is a polynomial, we have a stronger result. We can actually prove that all $T_k$ are polynomials for $k = 1, \ldots, 2s-1$. The proof is slightly involved and can be found in the appendix.

**Lemma 2** If $T_{2s}(u)$ is also a polynomial, $P(u)$ is a rational function and is regular at Type-II poles.

*Proof.* $T_{2s}(u)$ can be written in terms $P(u)$ and $Q(u)$ as

$$T_{2s}(u) = (D^{2s}P)^{++}(D^{2s}Q)^{--} - (D^{2s}P)^{--}(D^{2s}Q)^{++}. \tag{87}$$

From which we have

$$T_{2s}(0) = (D^{2s}\bar{P})^{++}(D^{2s}\bar{Q})^{--} - (D^{2s}\bar{P})^{--}(D^{2s}\bar{Q})^{++}, \tag{88}$$

where $D^n\bar{F}$ is defined as the value of $D^nF(u)$ at $u = 0$. From the definition of $D$, it is straightforward to derive that

$$D^nF(u) = \sum_{k=0}^{n}(-1)^k\binom{n}{k}F\left(u+\tfrac{i}{2}(n-2k)\right), \tag{89}$$

and therefore

$$D^n\bar{F} = \sum_{k=0}^{n}(-1)^k\binom{n}{k}F\left(\tfrac{i}{2}(n-2k)\right). \tag{90}$$

Plugging into (88), we find that on the right-hand side of (88), $(D^{2s}\bar{P})^{++}$ is finite due to Lemma 1. $(D^{2s}\bar{Q})^{++}$ and $(D^{2s}\bar{Q})^{--}$ are also finite because $Q(u)$ is a polynomial. The only potential divergence is contained in

$$(D^{2s}\bar{P})^{--} = \sum_{k=0}^{2s}(-1)^k\binom{2s}{k}P(i(s-k-1)) = (-1)^{2s}P(-is-i) + \texttt{finite terms}, \tag{91}$$

where the 'finite terms' are finite due to **Lemma 1**. By our assumption, $T_{2s}(0)$ is finite, it then follows that $P(-is-i)$ is also finite. Knowing that $P(-is)$ and $P(-is-i)$ are finite, we can again use the recursion relation (86) for $k = 2s+1, 2s+2, \ldots$ to prove that all $P(i(s-k-1))$ are finite for $k \geq 2s+1$. This proves that $P(u)$ is regular at Type-II poles.

To show that $P(u)$ is a rational function, notice that we can write $P(u)$ in the form

$$P(u) = P_0(u) + Q(u)\sum_{m=0}^{M-1}\sum_{j=-(s-1)}^{s} a_j^{(m)}\psi^{(m)}(-iu+j). \tag{92}$$

This can always be done by formally adding some terms with zero coefficients to (75) and (72) so that the upper limit of the sum over $m$ for each $j$ is the same. Using the infinite series representation of polygamma functions

$$
\psi^{(m)}(z) = \begin{cases} -\gamma + \sum\limits_{n=0}^{\infty}\left(\dfrac{1}{n+1} - \dfrac{1}{n+z}\right), & m = 0, \\[2ex] (-1)^{m+1}m!\sum\limits_{k=0}^{\infty}\dfrac{1}{(z+k)^{m+1}}, & m = 1, 2, \dots \end{cases}
\tag{93}
$$

we find that

$$
\psi^{(m)}(\epsilon + n) = \frac{(-1)^{m+1}m!}{\epsilon^{m+1}} + \cdots, \qquad \forall n = 0, -1, -2, \dots,
\tag{94}
$$

for $\epsilon \to 0$ where the ellipsis denotes regular terms. As a result, at $u = -is - in$ with $n > 0$, we have

$$
P(-i(s+n)) = P_0(-i(s+n)) + Q(-i(s+n))\Psi_\epsilon + \texttt{finite terms},
\tag{95}
$$

where $\Psi_\epsilon$ is defined as

$$
\Psi_\epsilon = \sum_{m=0}^{M-1}\left(\sum_{j=-(s-1)}^{s} a_j^{(m)}\right)\frac{(-1)^{m+1}m!}{\epsilon^{m+1}},
\tag{96}
$$

with $\epsilon \to 0$. We have proven that $P(u)$ is regular at type-II poles if $T_{2s}$ is a polynomial. At the same time, we know that $Q(u)$ is non-zero at type-II poles due to the **Proposition** in Appendix A.1. It then follows from (95) that $\Psi_\epsilon$ must vanish. This is equivalent to the conditions

$$
\sum_{j=-(s-1)}^{s} a_j^{(m)} = 0,
\tag{97}
$$

for all $m$. Plugging (97) into (92) and using the following property of the polygamma function

$$
\psi^{(m)}(z+1) - \psi^{(m)}(z) = \frac{(-1)^{m+1}m!}{z^{m+1}},
\tag{98}
$$

we see that $P(u)$ must be a rational function.

From **Lemma 1** and **Lemma 2**, if $T_0/\alpha$ and $T_{2s}$ are polynomials, $P(u)$ must be a polynomial. On the other hand, under the assumption that $T_0(u)/\alpha$ is a polynomial, $T_{2s}(u)$ being a polynomial is equivalent to $P(u)$ being a polynomial. Therefore **Theorem 2** is proven.

Since the polynomiality of $T_n$ is equivalent to the polynomiality of $Q_{0,n}$ via (27), **Theorem 2** has the following equivalent formulation

**Theorem 2'** $P(u)$ is a polynomial if $Q_{0,1}(u)/\alpha(u)$ and $Q_{0,2s+1}(u)$ are polynomials.

As mentioned before, this theorem encodes the *minimality condition* of the rational $Q$-system. In fact, we have shown that the essential condition is the polynomiality of $Q_{0,1}/\alpha$ and $Q_{0,2s+1}$, requiring the rest of $Q_{0,n}$ to be polynomial is redundant. This is consistent with the observation in the spin-$\frac{1}{2}$ case that we only need to consider the first two column (corresponding to $Q_{0,1}$ and $Q_{0,2}$) of the rational $Q$-system [12].

### 3.3 Completeness of rational $Q$-system

Based on the results of the previous subsection, we can prove the completeness of our rational $Q$-system based on the result of V. Tarasov [35]. Let us define the subspace of highest weight states in the $N$-magnon sector

$$\mathcal{H}_N = \left\{ |\psi\rangle \in \mathcal{H} \, | \, S^+|\psi\rangle = 0, \quad S^3|\psi\rangle = (sL - N)|\psi\rangle \right\}. \tag{99}$$

We quote **Theorem 6.2** of [35],[5] which states: *a polynomial $\tau(u)$ is an eigenvalue of the transfer matrix $T(u)$ on $\mathcal{H}_N$ if and only if the difference equation*

$$\tau(u)f(u) = (u + is)^L f(u - i) + (u - is)^L f(u + i), \tag{100}$$

*has two polynomial solutions $Q(u)$, $P(u)$ such that $\deg(Q) = N < \deg(P)$. Moreover, the corresponding eigenvector is unique up to proportionality. In particular, if $T(u)$ is diagonalizable on $\mathcal{H}_N$, then there exist exactly $\dim \mathcal{H}_N$ polynomials $\tau(u)$ such that (100) has polynomial solutions $Q(u)$, $P(u)$.*

From the results of the previous subsection, especially **Theorem 2'**, we see that our rational $Q$-system is equivalent to requiring that both solutions $Q(u)$ and $P(u)$ of the $TQ$-relation are polynomials. Thanks to the results in [35], the solutions of the rational $Q$-system give complete physical solutions of BAE. This is also supported by our numerical checks in the previous section.

## 4 Physical singular solutions

In this section, we derive physicality conditions for singular solutions from the rational $Q$-system, based on the assumption that the rational $Q$-system gives all physical solutions.

### 4.1 Physicality conditions

From the Wronskian relation, we have

$$\frac{P^+}{Q^+} - \frac{P^-}{Q^-} = \frac{q_\alpha}{Q^+ Q^-}, \tag{101}$$

from which follows

$$\frac{P^+}{Q^+} = \frac{P^{[-2n-1]}}{Q^{[-2n-1]}} + \sum_{k=0}^{n} \frac{q_\alpha^{[-2k]}}{Q^{[-2k+1]}Q^{[-2k-1]}}, \quad n = 0, 1, 2, \dots \tag{102}$$

Taking $u = u + i(s + \frac{1}{2})$ and $n = 2s + 1$ in the above relation, we obtain

$$\frac{P^{[2s+2]}}{Q^{[2s+2]}} - \frac{P^{[-2s-2]}}{Q^{[-2s-2]}} = \sum_{k=0}^{2s+1} \frac{q_\alpha^{[2s-2k+1]}}{Q^{[2s-2k+2]}Q^{[2s-2k]}}. \tag{103}$$

Multiplying both sides by $Q^{[2s+2]}Q^{[-2s-2]}$ leads to

$$P^{[2s+2]}Q^{[-2s-2]} - P^{[-2s-2]}Q^{[2s+2]} = \sum_{k=0}^{2s+1} \frac{Q^{[2s+2]}Q^{[-2s-2]}}{Q^{[2s-2k+2]}Q^{[2s-2k]}} q_\alpha^{[2s-2k+1]}. \tag{104}$$

---

[5]To be more precise, while the theorem works for general inhomogeneous spin chain, here we focus on the homogeneous limit. We also modified the notations slightly in order to be consistent with the current paper.

The left-hand side is regular at $u = 0$, therefore the right-hand should also be the case. Let us define

$$\mathbf{B}_s(u) \equiv \sum_{k=0}^{2s+1} \frac{Q^{[2s+2]}Q^{[-2s-2]}}{Q^{[2s-2k+2]}Q^{[2s-2k]}} q_\alpha^{[2s-2k+1]}. \tag{105}$$

We find that the physical condition for the singular solution can be compactly given by

$$\mathbf{B}_s(u) \quad \text{is regular at } u = 0. \tag{106}$$

This requirement might seem a bit strange at first glance. In fact, under the assumption that $T_0(u)/\alpha(u)$ is a polynomial, (106) is equivalent to the requirement that $\bar{P}^{[-2s-2]}$ is regular, which is equivalent to the polynomiality of $P(u)$, as can be seen from (104) and the proof of **Lemma 2**.

To test this proposal, let us consider a few examples.

**Example 1.** Consider the spin-$\frac{1}{2}$ case with $s = \frac{1}{2}$ and $q_\alpha(u) = u^L$. The $Q$-polynomial takes the form

$$Q(u) = \left(u - \tfrac{i}{2}\right)\left(u + \tfrac{i}{2}\right) \mathcal{Q}(u), \tag{107}$$

and we have

$$\begin{aligned}
\mathbf{B}_{\frac{1}{2}}(u) &= \frac{Q^{[3]}}{Q^-}(u-i)^L + \frac{Q^{[-3]}}{Q^+}(u+i)^L + \frac{Q^{[3]}Q^{[-3]}}{Q^+Q^-}u^L \\
&= \frac{(u-i)^{L-1}(u+i)(u+2i)\mathcal{Q}^{[3]}}{u\,\mathcal{Q}^-} + \frac{(u+i)^{L-1}(u-i)(u-2i)\mathcal{Q}^{[-3]}}{u\,\mathcal{Q}^+} \\
&\quad + u^{L-2}(u-2i)(u+2i)\frac{\mathcal{Q}^{[-3]}\mathcal{Q}^{[3]}}{\mathcal{Q}^-\mathcal{Q}^+}.
\end{aligned} \tag{108}$$

For spin chains with $L \geq 2$, which we assume here, the last term is regular at $u = 0$. There are spurious poles at $u = 0$ in the first two terms on the right-hand side of (108). We have

$$\operatorname*{Res}_{u=0} \mathbf{B}_{\frac{1}{2}}(u) = -2(-i)^{L-1}\frac{\mathcal{Q}^{[3]}}{\mathcal{Q}^-} - 2(i)^{L-1}\frac{\mathcal{Q}^{[-3]}}{\mathcal{Q}^+}. \tag{109}$$

Requiring the residue to vanish leads to the condition

$$\frac{\mathcal{Q}^{[3]}}{\mathcal{Q}^{[-3]}}\frac{\mathcal{Q}^+}{\mathcal{Q}^-} = \prod_{j=1}^{M_r}\left(\frac{u_j + \frac{3i}{2}}{u_j - \frac{3i}{2}}\right)\left(\frac{u_j + \frac{i}{2}}{u_j - \frac{i}{2}}\right) = (-1)^L, \tag{110}$$

which is precisely the physicality condition for singular solutions in the spin-$\frac{1}{2}$. Notice that our derivation is different from the one presented in [12].

**Example 2** We consider the spin-$s$ singular solution which contains exactly one length-$2s+1$ Bethe string $\{is, i(s-1), \ldots, -is\}$. Namely, there are no repeated singular roots. In this case, we have

$$Q(u) = (u - is)(u - i(s-1))\ldots(u + is) \mathcal{Q}(u), \tag{111}$$

and

$$\mathbf{B}_s(u) = \frac{Q^{[-2s-2]}}{Q^{[2s]}} q_\alpha^{[2s+1]} + \frac{Q^{[2s+2]}}{Q^{[-2s]}} q_\alpha^{[-2s-1]} + \cdots, \tag{112}$$

where the ellipsis denotes terms that are regular at $u = 0$. Plugging in $Q(u)$ and $q_\alpha(u)$, the residue is given by

$$\operatorname*{Res}_{u=0} \mathbf{B}_s(u) = i(2s+1)(-i)^{2s}[(2s)!]^L \left( \frac{\mathcal{Q}(i(s+1))}{\mathcal{Q}(-is)}(-i)^{2sL} - \frac{\mathcal{Q}(-i(s+1))}{\mathcal{Q}(is)}i^{2sL} \right). \tag{113}$$

We see that the requirement that the residue vanishes leads to

$$\frac{\mathcal{Q}(i(s+1))}{\mathcal{Q}(-i(s+1))} \frac{\mathcal{Q}(is)}{\mathcal{Q}(-is)} = (-1)^{2sL}. \tag{114}$$

This is a generalization of (110), which is equivalent to (9) after taking into account the rest BAE and matches the result from [27].

**Example 3** In this example we consider $s = 1$ with repeated singular roots. We have

$$Q(u) = (u-i)^{m_0} u^{m_1}(u+i)^{m_2}\, \mathcal{Q}(u), \quad m_i \geq 1. \tag{115}$$

$\mathbf{B}_s(u)$ is given by

$$\mathbf{B}_s(u) = (u-3i)^{m_0}(u-2i)^{m_1}(u-i)^{m_2}(u+i)^{L-m_1}(u+2i)^{L-m_2} \frac{\mathcal{Q}(u-2i)}{u^{m_0}\mathcal{Q}(u+i)} \tag{116}$$

$$+ (u-2i)^{L-m_0}(u-i)^{L-m_1}(u+i)^{m_0}(u+2i)^{m_1}(u+3i)^{m_2} \frac{\mathcal{Q}(u+2i)}{u^{m_2}\mathcal{Q}(u-i)} + \cdots,$$

where the ellipsis denotes two terms which are regular at $u = 0$. We can see the change due to repeated roots. If $m_0, m_2 > 1$, there are higher order poles at $u = 0$ and we need to impose more than one condition to ensure that $\mathbf{B}_s(u)$ is regular.

At the same time, since it contains repeated roots, we also need to take into account possible extra conditions from the polynomiality of $Q_{0,1}/\alpha$, which is equivalent to the polynomiality of $\tau(u)$. In our case,

$$\tau(u) = (u+i)^L \frac{Q(u-i)}{Q(u)} + (u-i)^L \frac{Q(u+i)}{Q(u)} \tag{117}$$

$$= (u-2i)^{m_0}(u-i)^{m_1-m_0}(u+i)^{L-m_2} \frac{\mathcal{Q}(u-i)}{u^{m_1-m_2}\mathcal{Q}(u)}$$

$$+ (u-i)^{L-m_0}(u+i)^{m_1-m_2}(u+2i)^{m_2} \frac{\mathcal{Q}(u+i)}{u^{m_1-m_0}\mathcal{Q}(u)}.$$

We see that if $m_1 > m_0, m_2$, there are extra conditions coming from the requirement

$$\operatorname*{Res}_{u=0} \tau(u) = 0. \tag{118}$$

Let us now consider the case $m_0 = 1, m_1 = 2, m_2 = 1$. From the regularity of $\mathbf{B}_1(u)$, we do not have extra conditions. Equation (114) specified at $s = 1$ gives

$$\frac{\mathcal{Q}(2i)}{\mathcal{Q}(-2i)} \frac{\mathcal{Q}(i)}{\mathcal{Q}(-i)} = 1. \tag{119}$$

From the regularity of $\tau(u)$ at $u = 0$, we have an extra condition, which reads

$$\frac{\mathcal{Q}(i)}{\mathcal{Q}(-i)} = (-1)^L. \tag{120}$$

Combining (119) and (120), we find the singular solutions containing $\{-i, 0, 0, i\}$ are physical if the regular roots satisfy the following two constraints

$$\prod_{k=1}^{M_r}\left(\frac{u_k+i}{u_k-i}\right)=(-1)^L, \qquad \prod_{k=1}^{M_r}\left(\frac{u_k+2i}{u_k-2i}\right)=(-1)^L. \tag{121}$$

The second constraint matches Equation (B16) of [27].

To summarize, when repeated singular roots occur, we need to impose regularity condition for both $\tau(u)$ and $\mathbf{B}_s(u)$ at $u=0$. We have seen that, depending on the multiplicities, the conditions can be different. It is unlikely that any choice of the multiplicities have solutions. Therefore it is an interesting and important question for given $L, M, s$, what the possible values of the multiplicities are. We will derive some of the constraints in the next subsection.

## 4.2 Constraints for multiplicities

In this subsection, we derive constraints for the possible values of $\{m_0, \ldots, m_{2s}\}$ for given $L$. These constraints are derived from a careful analysis of the $TQ$-relation at the singular roots $\{is, \ldots, -is\}$. Recall the $TQ$-relation reads

$$\tau(u)Q(u) = (u+is)^L Q(u-i) + (u-is)^L Q(u+i). \tag{122}$$

We consider the singular solution

$$Q(u) = (u-is)^{m_0}(u-i(s-1))^{m_1}\ldots(u+is)^{m_{2s}}\,\mathcal{Q}(u). \tag{123}$$

Taking $u = is + \epsilon$ in the $TQ$-relation, focusing on the leading power of $\epsilon$ on each term, we obtain

$$t_0\,\epsilon^{m_0} = \alpha_0\,\epsilon^{m_1} + \beta_0\,\epsilon^L, \tag{124}$$

where $t_0, \alpha_0, \beta_0$ are some constants and $\alpha_0, \beta_0 \neq 0$. Taking $u = i(s-j) + \epsilon$ for $j = 1, \ldots, 2s$, we will obtain similar relations, which can be written collectively as

$$t_j\,\epsilon^{m_j} = \alpha_j\,\epsilon^{m_{j+1}} + \beta_j\,\epsilon^{m_{j-1}}, \quad j = 0, 1, \ldots, 2s, \tag{125}$$

where we have defined $m_{-1} = m_{2s+1} = L$. In what follows, it is important to use the fact that $\alpha_j, \beta_j \neq 0$. We shall rule out the 'forbidden zone' for the values of $m_{j-1}, m_j, m_{j+1}$. Dividing both sides by $\epsilon^{m_{j+1}}$ in (125), we find

$$t_j\,\epsilon^{m_j-m_{j+1}} - \beta_j\,\epsilon^{m_{j-1}-m_{j+1}} = \alpha_j. \tag{126}$$

Now in the limit $\epsilon \to 0$, if $m_{j+1} < m_j$ and $m_{j+1} < m_{j-1}$, the left-hand side is zero while the right-hand side is a non-zero number. This is clearly a contradiction which shows that $(m_{j+1} < m_j) \wedge (m_{j+1} < m_{j-1})$ is forbidden. Similarly, we can prove that $(m_{j-1} < m_j) \wedge (m_{j-1} < m_{j+1})$ is also not allowed. In terms of words, it means neither $m_{j-1}$ nor $m_{j+1}$ can be strictly smaller than the rest two values in the tuple $\{m_{j-1}, m_j, m_{j+1}\}$.

It is possible to have $m_j$ strictly larger than the rest two. In this case, the first term in (126) $t_j\epsilon^{m_j-m_{j+1}} \to 0$. It then follows that we must have $m_{j-1} = m_{j+1}$, otherwise, the second term $\beta_j\,\epsilon^{m_{j-1}-m_{j+1}}$ either goes to zero or diverges in the limit $\epsilon \to 0$, which is inconsistent with the right-hand side.

Furthermore, let us denote

$$\mathfrak{m} = \min\{m_0, m_1, \ldots, m_{2s}\}. \tag{127}$$

Concerning $\mathfrak{m}$, we have the following two results

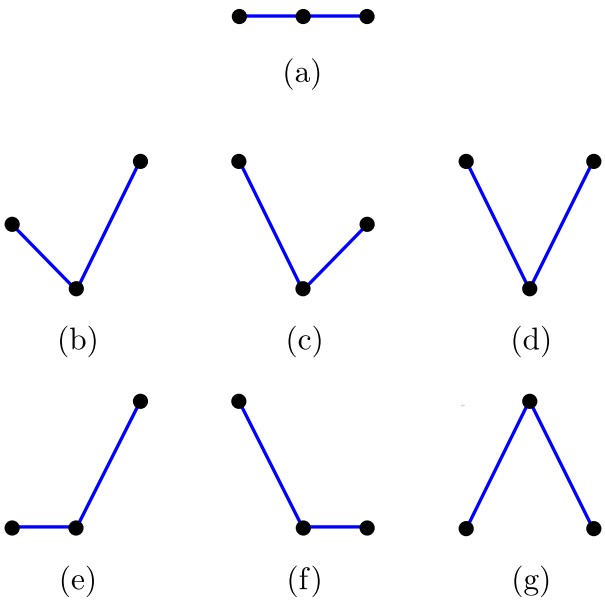

Figure 4: Possible configurations for multiplicities.

- $2\mathfrak{m} < L$

  *Proof.* We have

$$sL \geq M = \sum_{j=0}^{2s} m_j + M_r \geq (2s+1)\mathfrak{m}, \tag{128}$$

which implies that

$$L \geq (2 + s^{-1})\mathfrak{m} > 2\mathfrak{m}. \tag{129}$$

- $m_0 = m_{2s} = \mathfrak{m}$.

  *Proof.* Suppose $m_0 > \mathfrak{m}$. Consider the tuple $(L, m_0, m_1)$. It is clear that $L > \mathfrak{m}$. If $m_1 \leq \mathfrak{m}$, $m_1$ would be strictly smaller than the rest of the other two elements, which is against the forbidden rule. Therefore we must have $m_1 > \mathfrak{m}$.

  Now we consider the tuple $(m_0, m_1, m_2)$. We have $m_0 > \mathfrak{m}$ and $m_1 > \mathfrak{m}$. By a similar argument to the previous step, we conclude that we must have $m_2 > \mathfrak{m}$. Continuing in the same way, we conclude that $m_j > \mathfrak{m}$ for all $j = 0, 1, \ldots, 2s$. This is in contradiction with the assumption that $\mathfrak{m} = \min\{m_0, \ldots, m_{2s}\}$. Therefore, we must have $m_0 = \mathfrak{m}$. Similarly, we can prove that $m_{2s} = \mathfrak{m}$.

These considerations already give some constraints on the multiplicities. In order to visualize the possible values of the multiplicities, we can consider the following chain

$$(m_0, m_1, \ldots, m_{2s}), \tag{130}$$

and make a plot of their values. Our constraints imply that taking any three consecutive sites $m_{j-1}, m_j, m_{j+1}$, the possible configurations for the values can only be one of the 7 configurations in Figure 4. An example of the possible[6] values of the multiplicities is given in Figure 5. It is easy to see that the allowed configurations for multiplicities must take the shape that is formed by a number of 'peaks'.

Before ending this section, let us comment that the constraints we found here are still preliminary and might not be complete. It is interesting to deduce for given $L, M, s$, what the possible values of $m_0, \ldots, m_{2s}$ are. We wish to come back to this important point in the future.

---

[6]Here by 'possible' we only mean that this configuration is not against the forbidden rules which we derived.

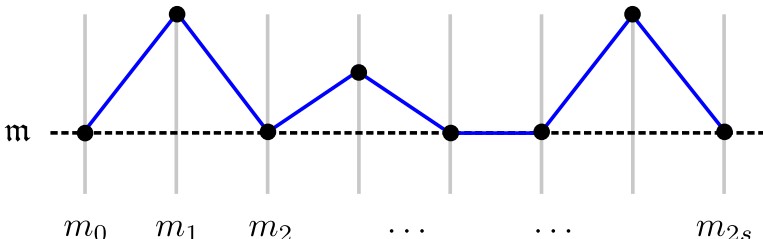

Figure 5: One possible configuration for the multiplicities.

# 5 Conclusions and outlook

In this paper, we proposed the rational $Q$-system for the XXX$_s$ spin chain. This $Q$-system is a natural generalization of the spin-$\frac{1}{2}$ case, but with important modifications. We tested numerically that the solution of the $Q$-system gives all and only physical solutions of the corresponding Bethe Ansatz equations.

We proved rigorously that the rational $Q$-system is equivalent to the requirement of polynomiality of $\tau(u)$, $Q(u)$, and the dual solution $P(u)$ of the $TQ$-relation. Based on this, we are able to prove the completeness of the proposed rational $Q$ system, and derive extra conditions for the singular solutions to be physical. In the spin-$s$ case, these constraints are much more involved and highly non-trivial for different multiplicities of the singular roots.

Finally, we initiated the investigation of the possible values of the multiplicities for the singular roots based on a careful analysis of $TQ$-relation at singular roots, which leads to a set of non-trivial constraints.

Our current work marks a solid step towards a deeper understanding of Bethe equations for spin-$s$ integrable spin chains, which is more difficult to analyze due to the presence of repeated roots. There are several important open questions and further directions to explore in the future.

Firstly, we have proposed the additional constraints for the singular roots based on our rational $Q$-system. It would be very desirable to derive these constraints directly from ABA with a proper regularization of the Bethe states and confirm that such constraints indeed coincide with physicality condition from the $Q$-system. Related to this question, it is important to classify all possible multiplicities of the singular roots $\{m_0, \ldots, m_{2s}\}$. We have only performed a preliminary analysis in the current work, a more systematic approach is called for.

At the same time, generalizations of the rational $Q$-system to other spin chains should also be considered. Some immediate cases include XXZ$_s$ chain with periodic and open boundary conditions, as well as their higher rank generalizations. In the XXX$_s$ case, we have proven that the rational $Q$-system is equivalent to the polynomiality of the $TQ$-relation and the dual solution. It is important to see whether this is still the case in the more general context.

Finally, let us mention that since the $Q$-system gives only physical solutions for the spin-$s$ chain, it can be applied to compute interesting quantities. Examples of this kind range from supersymmetric gauge theories [16, 36, 37] to statistical mechanics [38–40]. In particular in the context of the Bethe/gauge correspondence [41, 42], it is natural to consider XXX$_s$ spin chains as the dual integrable models to the generic 2d supersymmetric gauge theories, but most of the relevant studies have been focusing on the case dual to spin-$\frac{1}{2}$. For example, the Bethe wavefunction and the $R$-matrix have been only constructed for XXX$_{\frac{1}{2}}$ chain and its nested version from the gauge theory side [43, 44]. The rational $Q$-system proposed for XXX$_s$ spin chains in this article may help the future study of more general cases, *e.g.* singular loci appearing in supersymmetric gauge theories with a special FI parameter and $\theta$-angle [45] might be understood better with such powerful tools developed in the dual Bethe side.

# Acknowledgments

We would like to thank Yichao Liu, Hongfei Shu, Zixi Tan, Peng Zhao and Hao Zou for related discussions.

**Funding information** J.H. is supported by the Jiangsu Funding Program for Excellent Postdoctoral Talent. Y.J. is supported by the Startup Funding no. 3207022217A1 of Southeast University. R.Z. is supported by National Natural Science Foundation of China No. 12105198 and the High-level personnel project of Jiangsu Province (JSSCBS20210709).

# A Proof of some results

In this appendix, we give the proofs for two claims in the main text.

## A.1 Singular roots

For the spin-$s$ BAE

$$\left(\frac{u_j - is}{u_j + is}\right)^L = -\prod_{k=1}^{M}\frac{u_j - u_k - i}{u_j - u_k + i}, \tag{A.1}$$

we have the following result for the singular roots.

**Proposition 1** Suppose $\mathtt{sol} = \{u_k | k = 1, 2, ..., M\}$ is a solution of BAE (A.1). $\exists u_1, u_2 \in \mathtt{sol}$, such that $u_1 - u_2 = i$, if and only if $\mathtt{sol} \supset \{-is, -i(s-1), ..., is\}$.

*Proof.* We start by analyzing the BAE for $u_1$. If $u_1 \neq is$, the left-hand side of (A.1) is non-zero. In order for the right-hand side of (A.1) to be non-zero, there must exist an element in $\mathtt{sol}$, which without loss of generality is denoted by $u_3$, such that $u_1 - u_3 = -i$. We then analyze the BAE of $u_3$ in a similar way and conclude that if $u_3 \neq is$, there must exist a $u_5$ such that $u_3 - u_5 = -i$ and so on and so forth.

Similarly, for the BAE of $u_2$, if $u_2 \neq -is$, there $\exists u_4 \in \mathtt{sol}$, such that $u_2 - u_4 = i$. We then consider the the BAE of $u_4$. In this way, we get a chain of roots

$$\{\ldots, u_{2n}, \ldots, u_6, u_4, u_2, u_1, u_3, u_5, \ldots, u_{2m+1}, \ldots\} \subset \mathtt{sol}, \tag{A.2}$$

where the difference between each two adjacent elements is $-i$. Because the number of elements in $\mathtt{sol}$ is equal to $M < \infty$, the length of the chain must be less than or equal to $M$. For the chain to truncate, the must $\exists u_{2n}, u_{2m+1}$ such that $u_{2n} = -is$ and $u_{2m+1} = is$, that is, $\mathtt{sol} \supset \{-is, -i(s-1), ..., is\}$. For $u_1 = is$ or $u_2 = -is$, the proof is similar.

One important consequence of this proposition that we used in the main text is that $u = \pm i(s+1)$ cannot be contained in the solution if $\mathtt{sol} \supset \{-is, -i(s-1), ..., is\}$. Otherwise by analyzing the BAE for $u = \pm i(s+1)$, we would find the chain of roots (A.2) do not truncate and contain infinitely many elements, which is in contradiction to the fact that we have $M$ Bethe roots.

## A.2 Polynomiality of $T_n$ for $n < 2s$

In this appendix, we prove that if $T_0/\alpha$ is a polynomial, then in fact $T_n$ with $n < 2s$ are polynomials. It is useful to introduce the following quantities

$$S_n \equiv P^{[n]}Q^{[-n]} - P^{[-n]}Q^{[n]}, \qquad n = 1, 2, \ldots \tag{A.3}$$

In particular, we have

$$S_1(u) = q_\alpha(u), \qquad S_2(u) = T_0(u). \tag{A.4}$$

We have the following lemma.

**Lemma 3**  $T_n$ can be written as a linear combination of $q_\alpha, T_0, T_1, \ldots, T_{n-1}$ and $S_{n+2}$ as

$$T_n = (-1)^n S_{n+2} + \sum_k c_{\alpha,k} q_\alpha^{[2k]} + \sum_{j=1}^{n-1} \sum_k c_{j,k} T_j^{[2k]}, \tag{A.5}$$

where $c_{\alpha,k}, c_{j,k}$ are constants and $2k \in \mathbb{Z}$. These constants can be worked out explicitly, but are not relevant to our proof.

*Proof* We write

$$D^n P = \sum_{k=0}^n c_k P^{[n-2k]}, \qquad D^n Q = \sum_{j=0}^n d_j Q^{[n-2j]}, \tag{A.6}$$

where $c_k, d_j$ are constants. In particular, $c_0 = d_0 = 1$ and $c_n = d_n = (-1)^n$. We then have

$$(D^n P)^{++} (D^n Q)^{--} = (-1)^n P^{[n+2]} Q^{[-n-2]} + \sum_{(k,j) \neq (0,n)} c_k d_j P^{[n-2k+2]} Q^{[n-2j-2]}. \tag{A.7}$$

Noticing that in the sum each term can be written as

$$P^{[n-2k+2]} Q^{[n-2j-2]} = \left( P^{[m]} Q^{[-m]} \right)^{[2k]}, \tag{A.8}$$

with

$$n - 2k + 2 = m + 2k, \qquad n - 2j - 2 = -m + 2k. \tag{A.9}$$

It is then obvious that we can write

$$T_n = (D^n P)^{++} (D^n Q)^{--} - (D^n P)^{--} (D^n Q)^{++} \tag{A.10}$$

$$= (-1)^n S_{n+2} + \sum_{j=1}^{n+1} \sum_k \tilde{c}_{j,k} S_j^{[2k]},$$

where $\tilde{c}_{j,k}$ are constants. This tells us that $T_n$ can be written as a linear combination of $S_1, S_2, \ldots, S_{n+2}$, with proper shifts in the spectral parameter. Using (A.4) as initial conditions, we can solve the linear system recursively and write $S_n$ as a linear combination of $T_k$ and $q_\alpha$. This proves (A.5). Now we can prove the main result.

**Proposition 2**  If $T_0/\alpha$ is a polynomial, $T_n$ with $n < 2s$ are all polynomials.

*Proof* We shall prove the result by induction. According to **Lemma 3** we just proved. What we need to show is that if $T_0/\alpha$ and $S_3, S_4, \ldots, S_{n+1}$ are polynomials, then $S_{n+2}$ is also a polynomial for all $n < 2s$.

Plugging in the explicit form of $P(u)$ into the definition of $S_n$, we obtain

$$S_{n+2} = P_0^{[n+2]} Q^{[-n-2]} - P_0^{[-n-2]} Q^{[n+2]} \tag{A.11}$$

$$+ Q^{[n+2]} Q^{[-n-2]} \sum_{m=0}^{M-1} \sum_{j=-(s-1)}^{s} a_j^{(m)} \left[ \psi^{(m)}(-iu + j + \tfrac{n+2}{2}) - \psi^{(m)}(-iu + j - \tfrac{n+2}{2}) \right].$$

Therefore, the potential poles of $S_{n+2}$ from polygamma functions are

$$\left\{-i(s+\tfrac{n}{2}), -i(s+\tfrac{n}{2}-1), \ldots, i(s+\tfrac{n}{2})\right\}. \tag{A.12}$$

From **Lemma 3**, we find that if $T_0/\alpha$, $S_3, S_4, \ldots, S_{n+1}$ are polynomials, $T_n$ and $S_{n+2}$ have the same potential poles. It is also clear from (A.11) that $S_{n+2}(u)$ is a rational function. So if we can prove that $S_{n+2}$ are regular at the potential poles, then $S_{n+2}$ must be a polynomial.

Let us first prove that $S_3(u)$ is a polynomial. Taking $u$ at the value of the potential poles, we obtain

$$\bar{S}_3^{[-2s-1]} = \bar{P}^{[-2s+2]}\bar{Q}^{[-2s-4]} - \bar{P}^{[-2s-4]}\bar{Q}^{[-2s+2]}, \tag{A.13}$$
$$\bar{S}_3^{[-2s+1]} = \bar{P}^{[-2s+4]}\bar{Q}^{[-2s-2]} - \bar{P}^{[-2s-2]}\bar{Q}^{[-2s+4]},$$
$$\bar{S}_3^{[-2s+3]} = \bar{P}^{[-2s+6]}\bar{Q}^{[-2s]} - \bar{P}^{[-2s]}\bar{Q}^{[-2s+6]},$$
$$\bar{S}_3^{[-2s+5]} = \bar{P}^{[-2s+8]}\bar{Q}^{[-2s+2]} - \bar{P}^{[-2s+2]}\bar{Q}^{[-2s+8]},$$
$$\cdots \quad \cdots$$
$$\bar{S}_3^{[2s+1]} = \bar{P}^{[2s+4]}\bar{Q}^{[2s-2]} - \bar{P}^{[2s-2]}\bar{Q}^{[2s+4]}.$$

Since $P(u)$ is regular at Type-I poles by **Lemma 1**, $\bar{S}_3^{[n]}$ is regular at $n = -2s+3, -2s+5, \ldots, 2s+1$. $\bar{S}_3^{[-2s-1]}$ and $\bar{S}_3^{[-2s+1]}$ in (A.13) are special because the right-hand side involve $P(u)$ at Type-II poles (the two factors in blue color). On the other hand, $Q(u)$ have zeros at the same points. We shall show that the potential divergences in $P(u)$ are cancelled neatly by the corresponding zeros in $Q(u)$, leaving a finite result. To this end, recall the relations

$$T_n D^n Q = Q_{0,n}^+ D^n Q^{--} + Q_{0,n}^- D^n Q^{++}, \tag{A.14}$$

and

$$T_n = Q_{0,n}^+ + Q_{0,n}^- - Q_{0,n+1}. \tag{A.15}$$

Taking $n = 0$ in (A.15), we see that $Q_{0,1}$ is a polynomial since by assumption $T_0$ is a polynomial. Now taking $n = 1$ in (A.14),

$$T_1 DQ = Q_{0,1}^+ DQ^{--} + Q_{0,1}^- DQ^{++}. \tag{A.16}$$

The right-hand side is a polynomial, therefore $T_1 DQ$ is also a polynomial. This implies that the potential poles of $T_1$ must be cancelled by the zeros of $DQ$. We have

$$DQ = \left(u - i(s-\tfrac{1}{2})\right)^{m_0} \left(u - i(s-\tfrac{3}{2})\right)^{m_1} \ldots \left(u + i(s+\tfrac{1}{2})\right)^{m_{2s}} \mathcal{Q}^+ \tag{A.17}$$
$$- \left(u - i(s+\tfrac{1}{2})\right)^{m_0} \left(u - i(s-\tfrac{1}{2})\right)^{m_1} \ldots \left(u + i(s-\tfrac{1}{2})\right)^{m_{2s}} \mathcal{Q}^-,$$

from which we see that $u = -i(s+\tfrac{1}{2})$ cannot be the zero of $DQ$, otherwise this would imply that $-i(s+1)$ is a zero of $\mathcal{Q}(u)$, which is in contradiction with **Proposition 1**. This in turn implies that $T_1(u)$ cannot have a pole at $u = -i(s+\tfrac{1}{2})$. By (A.5), we conclude that $\bar{S}_3^{[-2s-1]}$ is regular.

Now we continue to show that $\bar{S}_3^{[-2s+1]}$ is regular. From the first equation of (A.13) and the fact that $\bar{S}_3^{[-2s-1]}$ is regular, it follows that $\bar{P}^{[-2s-4]}\bar{Q}^{[-2s+2]}$ is regular. Using (95), we conclude that $\bar{Q}^{[-2s+2]}\Psi_\epsilon$ is regular. This implies that $\epsilon^{m_{2s-1}}\Psi_\epsilon$ is regular. At the same time, we know that $\bar{P}^{[-2s]}$, which implies that $\bar{Q}^{[-2s]}\Psi_\epsilon$ and $\epsilon^{m_{2s}}\Psi_\epsilon$ are regular. We have proven in section 4.2 that neither $m_{2s}$ nor $m_{2s-2}$ can be strictly smaller than the rest two values in the tuple $\{m_{2s}, m_{2s-1}, m_{2s-2}\}$. This implies that $\epsilon^{m_{2s-2}}\Psi_\epsilon$ must also be regular, otherwise $m_{2s-2}$ would have to be smaller than both $m_{2s}$ and $m_{2s-1}$. Now $\epsilon^{m_{2s-2}}\Psi_\epsilon$ being regular is equivalent to $\bar{Q}^{[-2s+4]}\Psi_\epsilon$ being regular, which in turn implies that $\bar{S}_3^{[-2s+1]}$ is regular.

Similarly, we can prove that $\epsilon^{m_{2s-3}}\Psi_\epsilon,\ldots,\epsilon^{m_0}\Psi_\epsilon$ are regular, which are equivalent to $\bar{Q}^{[-2s+6]}\Psi_\epsilon,\ldots,\bar{Q}^{[2s]}\Psi_\epsilon$ being regular. These can be used to show that $S_4,\ldots,S_{2s+1}$ are regular at all the potential poles. Therefore $T_n$ with $n<2s$ are all polynomials.

Finally, let us make a remark. In fact, if $T_0/\alpha$ is a polynomial, we have shown that $\epsilon^{m_k}\Psi_\epsilon$ is regular for $k=0,1,\ldots,2s$. Therefore we obtain the upper bound of the summation over $m$ in $\Psi_\epsilon$. Namely, if we denote $\mathfrak{m}=\min\{m_0,m_1,\ldots,m_{2s}\}$, we have

$$\Psi_\epsilon = \sum_{m=0}^{\mathfrak{m}-1}\left(\sum_{j=-(s-1)}^{s} a_j^{(m)}\right)\frac{(-1)^{m+1}m!}{\epsilon^{m+1}}\,. \tag{A.18}$$

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
