# Peer review of "Spin-$s$ Rational $Q$-system"

_SciPost Physics, doi:SciPost Phys. 16, 113 (2024)_

## Round 2 · Referee Report · Anonymous (Referee 1) · 2023-8-31

Strengths
1. New and non-trivial way to distinguish physical solutions of the Bethe equations for higher spin chains.
2. Introduction of the $Q$ system for the higher spin chains
3. New simple condition for physical singular solutions (4.6)
Weaknesses
1. Some proofs are not clearly written and should be improved
2. Notations sometime leading to confusion
3. Twisted case is mentioned but not properly discussed
4. English grammar
Report
The results of the paper "spin-$s$ rational $Q$ system" are clearly interesting and important, it brings significant new understanding of the Bethe equations for the higher spin chains. In particular it gives a new not straightforward property to distinguish physical singular solutions (4.6) which is quite remarkable. The authors make the effort to produce detailed proofs of their results. In general I think that ultimately these results should be published in SciPost however there are some intermediate steps which should be modified therefore I recommend a major revision. There are several points which should be improved
The proof of the Theorem 1 is not clear at all, there some crucial points that should be clarified.
In (3.9) the function $q(u)$ (without any subscript) appears for the first time without being properly introduced before. It is not clear what is this function and where it comes from. As it plays the crucial role later it should be properly defined (as far as I understand to obtain (3.7) the functions $q_\pm$ should have the form (3.11) but it is never stated).
More important remark: it is not clear from the proof why the proposed construction of $P(u)$ gives a solution of the TQ relation. In my opinion the proof should be carefully rewritten.
The proof of the Theorem 2 is not clear too, there are some problems with notations. In particular, as far as I understand the notation $F^{[k]}$ means the shift of the argument while the bar $\bar{F}^{[k]}$ means fixing $u=0$. In (3.28) these two operations are introduced together and a reader needs some guesswork to reproduce the proof, specially as the authors use the word "regular" for $\bar{P}$ while it is a number and not a function (the word finite will be more suitable in my opinion).
The third major point is the twisted case which is mentioned throughout the paper without any proper consideration. It is not properly introduced, even the Bethe equations are not written and for all the future analysis it is not treated at all (consequences of the $SU(2)$ symmetry breaking, what happens with singular solutions, which theorems work for the twisted case and what should be modified etc.). In my opinion this case deserves a more detailed explanation if it is mentioned.
The last remark is that a careful proofreading is very much recommended.
Requested changes
Major points:
1. Rewrite the proof of the Theorem 1
2. Fix the notations $Q^{[n]}$
3. Explain more carefully the case of twisted boundary conditions
4. Improve the English grammar
Minor points:
5. Better explain the completeness conjecture (2.4), in particular what are strange solutions
6. The figure 2.2 is a little bit misleading as it gives an impression that the box $1,2s+1$ is the last one in the upper row.

---

## Round 2 · Referee Report · Anonymous (Referee 2) · 2023-11-2

Strengths
The manuscript addresses some interesting integrable models and provides for it new and relevant results toward the complete description of its spectrum by functional equations.
Weaknesses
Different technical points require to be clarified as well as a better description of the research framework in which this work is located.
Report
In this manuscript the authors study the homogeneous Heisenberg chain with spin s higher than 1/2. They do this analysis in a purely functional approach (i.e. without a direct construction of the eigenstates) based on the analysis of TQ-equations and QQ-equations, the so-called “spin-s rational Q-system”. Their aim is to show that this system of equations gives indeed all and only the physical solutions (i.e. the ones associated to true eigenvalues of the transfer matrix/Hamiltonian) and they in fact check it numerically for several cases. The main result is that the spin-s rational Q-system, here presented, is equivalent to the requirement that the T-function and the two associated Q-/P-solution are all polynomials for the periodic chain. This result allows the author to use Tarasov results on completeness for the spin s homogeneous XXX chains, thanks to which the completeness of the rational Q-system is derived in the periodic chain.
So, the manuscript addresses some interesting integrable models and provides for it new and relevant results toward the complete description of its spectrum by functional equations. This also allows the authors to introduce some easily formulated conditions that have to be satisfied by the physical solutions of the Bethe equations. This sheds some first light on the complete set of extra conditions to be satisfied by the physical singular solutions to the Bethe equations for the spin s cases; a priori a non-trivial task to obtain from other approaches like Algebraic Bethe Ansatz for these spin s cases in which one could have both repeated and singular physical solutions.
While I consider the manuscript relevant, I think that some modifications are required in order to publish it on SciPost.
Such modifications go from technical points to be clarified (with also better distinction between results proven and only verified) to a better description of the research framework in which this work is located.
Let me start pointing out that I agree with the first referee about the required clarifications about Theorem 1 and 2 that he has listed. About it I would add the following observations about P-function and Wronskian. Note that from the P-function and Q-function satisfying the TQ-equation with the same T-function, it is a trivial exercise to derive a first order difference equation for the Wronskian with coefficients the same coefficients of the TQ-equation. Then it is an easy exercise to prove that the Wronskian is given by equation (3.2), once it is a polynomial. As it is stated it is not clear if the proof of Theorem 1 uses this argument in the derivation of (3.2). Do the authors are supposing that the P-function in (3.22) is a solution of the TQ-equation as consequence of the fact that the Q-function is a solution and that the Wronskian formed out of P and Q is a solution of the Wronskian equation? In this case they should make this explicit. That is, they should write explicitly the first order Wronskian equation, they should comment on the fact that (3.23) is a solution of it and from it derive that the P-function (3.22) is solution of the TQ-equation.
The other central technical point is about the twisted case, as the first referee pointed out, this is not introduced at all. In fact, just the QQ-relations are modified by a twist parameter in (2.9), while this is an important case which deserve a careful analysis. So, the authors should give some references about it and methods that have been used to analyze it and they should write something about the twist matrix and twisted boundary conditions, are they considering diagonal or generic twist?
The authors have some numerical result about completeness in the twisted case, however, one should mention that in this case several steps in their spin-s rational Q-system approach are missing or even not working currently. The first missing point is about a possible derivation of completeness. Indeed, the Tarasov results for the homogeneous XXX spin s system (in reference [13]) has been derived in the case of periodic boundary conditions. Even more intrinsically, the very idea at the basis of the method, i.e. that the spin-s rational Q-system expresses the polynomiality of the two solutions P and Q of the TQ-equation associated to the same T-function is here missing. Indeed, it has been shown in the papers https://iopscience.iop.org/article/10.1088/1751-8113/49/10/104002, for spin 1/2, and https://scipost.org/10.21468/SciPostPhys.10.2.026, for higher spin s, that in the twisted case there exists only one polynomial solution to the TQ-equation for a fixed T-function. There, but for the inhomogeneous models, it is proven that in fact completeness follows just using this one polynomial Q-function for each T-function. In this inhomogeneous framework, it is also interesting to remark the simplicity on the conditions required on the Q-function roots to define physical solutions, see for example Theorem 5.1 of the second cited paper. This also brings to the natural question if the authors can with their approach start from an inhomogeneous model, get some general prescription on the physical Bethe roots and then see how to adapt them in the homogeneous limit.
A part the technical points to clarify, above listed, I think that the authors should do the effort to better introduce the integrability background and research that surround their work.
To gives some examples, let me mention that from the introduction of the Q-operator and TQ-equations since the seventies by Baxter, for which the authors only cite the Baxter book [10], there is a vast and variegate literature using these tools and their generalizations, in particular, to develop pure functionals approaches to study quantum integrable models which go far beyond the XXX and XXZ models even for higher spins. One clear example is the so-called Analytic Bethe Ansatz of Reshetikhin, already introduced in the eighties. I found bizarre the attitude of the authors that make a jump of more than 40 year from Baxter works to their references [11,12,14-17]. They should take into account some more literature in between, this should give them also the opportunity to point out that the use of these Q-systems and TQ-equation is not limited to these special representations, for example, the higher rank case and in the prospective to enlarge the use of the analysis to other interesting integrable models.
Even when the authors focus on works directly in their analysis this is not always done consistently. The main example is the paper of Tarasov [13], which is cited only in section 3, while this is just the central tool for their derivation of completeness in the periodic case that is claimed already in abstract; its role in their analysis should be cited already in the introduction.
Finally, some more proofreading is desirable, I spotted for example the repetition of article “the an” in the phrase between eq. (1.1)-(1.2).

---

## Round 3 · Referee Report · Anonymous · 2024-2-21

Report
The proofs of two main theorems were considerably improved in this new version. The new version is much better than the previous one I recommend to accept the paper after few remaining minor corrections (listed below)
Requested changes
1. There is a misprint in the proof of theorem 1 (eq. 3.24)
2. The statement of the theorem 2 is "if and only if" so even is the fact that if $P(u)$ is a polynomial $T_0/\alpha$ and $T_{2n}$ are polynomials too is straightforward and follows as far as I understand from (2.29) it should be stated explicitly in the proof (instead of the sentence "on the other hand under the assumption etc" which has no meaning).
3. Careful proofreading is still required

---

## Round 3 · Author Response

Reply to referee 1.
We would like to thank the referee for the careful work and positive comments on the results of our work. Below we provide answers for the points raised in the referee's report.
- "The proof of theorem 1 is not clear."
We have rewritten the proof of theorem 1. The main modifications are : (1) The typo in eq(3.9) in the previous version is corrected. (2) More explanations are added in order to make the proof more readable. (3) By the end of the proof of theorem 1, we have added a paragraph (from eq(3.23) to eq(3.27)) to prove that the $P(u)$ which has been constructed in the previous steps is indeed a solution of the $TQ$-relation with the same transfer matrix.
- "The proof of theorem 2 is not clear too, there are some problems with notations."
In order to eliminate confusion, we have changed the notations. All the $\bar{F}^{[k]}$ are replaced by the more explicit $F(ik/2)$. Following the suggestions by the referee, we use finite' instead of
regular' for all the quantities which are numbers.
- "The third major point is the twisted case which is mentioned throughout the paper with any proper consideration."
Indeed the mention of the twisted case is rather brief. In the revised version, we added a subsection (section 2.3) to provide a proper introduction to the twisted case. Here we only consider the so-called diagonal twist, which preserves U(1) symmetry. Non-diagonal twists need significant modifications. We give the construction of twisted rational $Q$-system and checked that the $Q$-system indeed give all and only physical solutions also for the diagonal twist case. Note that, however, the proof of completeness for the periodic case for the rest of the paper cannot be applied directly to the twisted case. Therefore we prefer to give a more detailed discussion on the twisted case in a separated paper.
- We also did a more careful proofread.
We hope these modifications are sufficient to address the questions and comments from the referee.
Reply to referee 2.
We thank the referee for the careful work and the detailed comments and suggestions. Below we provide our answers to the points that are raised in the referee's report.
1.Clarification of theorem 1 and 2
We have made significant modifications for the proof of theorem 1 and 2. More specific to the referee's comment, in theorem 1 what we prove is the following: From the polynomial solution $Q(u)$ of $TQ$-relation, we can construct a function called $P(u)$ which satisfies the Wronskian equation eq(3.2). It then follows that the function $P(u)$ is the other solution of the $TQ$-relation. We have made this more explicit by changing the wording of theorem 1. We also added a paragraph at the end of proof of theorem 1 to show that $P(u)$, which satisfies the Wronskian equation is the other solution of the $TQ$-relation.
- About twisted case
Indeed the mention of the twisted case was rather brief in the draft. In the revised version, we have added a subsection (section 2.3 in the current version) to give a more proper introduction to the twisted case. Here we consider the diagonal twist, which preserves U(1) symmetry. The rational $Q$-systems for models with non-diagonal twist would require more significant modifications. For the diagonal twist, we present the construction of the rational $Q$-system, which amounts to a slight modification of the $QQ$-relation. We checked numerically that the proposed $Q$-system give all and only physical solutions of the BAE. More comments on the numerical results are added at the end of in section 2.5.
On the other hand, to prove that the proposed $Q$-system gives complete physical solutions of BAE is a different story. Indeed our proof for the periodic case which require both solutions of $TQ$-relation are polynomials cannot be adapted directly to the twisted case. As the referee has correctly pointed out, there exist only one polynomial solution for the $TQ$-function with fixed $T$. The inhomogeneous case has been discussed in the two papers mentioned by the referee, which are also cited in the revised version. Therefore we feel it is proper to give a more detailed discussion on the twisted case in a separate work which will appear soon.
- About references
The referee suggests that we take into account more references on $TQ$-relations and $Q$-system. We are of course aware of and fully appreciate the fact that the applications of $TQ$-relations and $Q$-systems go way beyond the current context. It is precisely because of this, taking into account all works in this direction becomes a daunting task. In the current work, our main focus is on solving Bethe ansatz equations. We take the perspectives that $TQ$-relation and rational $Q$-system are reformulations of the BAE of the Heisenberg spin chain. Also we want to mention that the rational $Q$-system proposed by Marboe and Volin is related to, but different from the traditional $Q$-system. Therefore we mainly cite the papers on the rational $Q$-system and gave the referee an impression that we `make a jump of more than 40 years'. At the same time, we also understand the referee's concern and we added more references on the $TQ$-relations and $Q$-systems in the second paragraph of the Introduction. As we mentioned before, this cannot be the complete list but we feel these are the most relevant ones. If there are other papers which the referee believes is worth citing, please let us know.
- In addition to the previous modifications, we also did a more careful proofread.
We hope these modifications are sufficient to address the questions and comments from the referee.

---

## Round 3 · List of Changes

List of changes:
1. The proof of Theorem 1 is rewritten. The statement is made more precise, more explanations are added in the proof; A paragraph is added to prove that $P(u)$ is the solution of the same $TQ$-relation.
2. Modification of the proof of Theorem 2. Changed to more explicit and less confusing notations.
3. A subsection (section 2.3) is added to introduce the case with diagonal twisted boundary condition. More discussions on numerics are added in section 2.5.
4. More references on TQ- and QQ-relations are added in the second paragraph of the introduction.
5. Some typos are corrected.

---

## Editorial Decision

published